# Determining subunit-subunit interaction from statistics of cryo-EM images: observation of nearest-neighbor coupling in a circadian clock protein complex

Xu Han[1], Dongliang Zhang[1], Lu Hong[2], Daqi Yu[1], Zhaolong Wu[1], Tian Yang[1], Michael Rust[3] ✉, Yuhai Tu[4] ✉ & Qi Ouyang[1,5] ✉

Biological processes are typically actuated by dynamic multi-subunit molecular complexes. However, interactions between subunits, which govern the functions of these complexes, are hard to measure directly. Here, we develop a general approach combining cryo-EM imaging technology and statistical modeling and apply it to study the hexameric clock protein KaiC in Cyanobacteria. By clustering millions of KaiC monomer images, we identify two major conformational states of KaiC monomers. We then classify the conformational states of (>160,000) KaiC hexamers by the thirteen distinct spatial arrangements of these two subunit states in the hexamer ring. We find that distributions of the thirteen hexamer conformational patterns for two KaiC phosphorylation mutants can be fitted quantitatively by an Ising model, which reveals a significant cooperativity between neighboring subunits with phosphorylation shifting the probability of subunit conformation. Our results show that a KaiC hexamer can respond in a switch-like manner to changes in its phosphorylation level.

Circadian clocks are endogenous biological processes that exhibit self-sustained oscillations with an approximately 24 h period[1,2]. These rhythms are found across diverse organisms from prokaryotes to eukaryotes[3,4]. In animals, disruption of circadian clock function causes temporal disorganization of physiology and contributes to a variety of diseases, such as disorders of the nervous system, cancer, and cardiovascular and cerebrovascular diseases[5-9].

Cyanobacteria are the simplest organisms known to possess a circadian clock[10,11]. Because the cyanobacterial oscillator can be reconstituted in vitro using three proteins, KaiA, KaiB, and KaiC[12], it presents an opportunity to uncover fundamental biophysical principles of circadian timing. This oscillator is realized by the interactions and conformational changes among the three Kai proteins, and the oscillation is manifest as rhythmic phosphorylation of KaiC[12,13].

The key enzyme in this circadian clock is KaiC. Many features of this protein have been revealed, largely from bulk studies of activity and high resolution structures based on averaging over many hexameric particles. KaiC forms a homo-hexamer complex with a "double doughnut" shape where each KaiC monomer (subunit) consists of two domains (N-terminal CI domain and C-terminal CII domain), with a total of 12 nucleotides bound to each hexameric particle[14,15]. Both the CI domain and CII domain have ATPase activity, i.e., the ability to catalyze the hydrolysis of adenosine triphosphate (ATP)[16-21].

[1]State Key Laboratory of Artificial Microstructure and Mesoscopic Physics, School of Physics, Peking University, Beijing 100871, China. [2]Graduate Program in Biophysical Sciences, University of Chicago, Chicago, IL 60637, USA. [3]Departments of Molecular Genetics and Cell Biology and of Physics, University of Chicago, Chicago, IL 60637, USA. [4]IBM T. J. Watson Research Center, Yorktown Heights, NY 10598, USA. [5]Center for Quantitative Biology and Peking-Tsinghua Center for Life Sciences, AAIC, Peking University, Beijing 100871, China. ✉e-mail: mrust@uchicago.edu; yuhai@us.ibm.com; qi@pku.edu.cn

Each KaiC monomer has two observed phosphorylation sites in the CII domain, Ser431 (S) and Thr432 (T), whose phosphorylation and dephosphorylation follow a cyclic order: ST→ SpT→ pSpT→ pST→ ST[22,23], where pT and pS represent the phosphorylated T and S residue, respectively. This phosphorylation-dephosphorylation cycle forms the basis of the 24 h circadian rhythm. During the first twelve hours of the cycle (day phase), KaiA stimulates phosphorylation by binding to the KaiC C-terminal tail and remodeling the so called A-loop domain of the KaiC protein (residue 488–497), allowing nucleotide exchange for phosphorylation of KaiC[19,24–29], i.e., ST→ SpT→ pSpT. This phosphorylation process is terminated in the next twelve hours of the cycle (night phase) when KaiB binds to phosphorylated KaiC and sequesters KaiA, allowing KaiC to dephosphorylate[30–35], i.e., pSpT→ pST→ ST.

While this phosphorylation-dephosphorylation cycle describes changes that occur to an individual KaiC monomer, for the entire KaiC hexamer to oscillate coherently, individual KaiC monomers in a hexamer need to coordinate their phosphorylation-dephosphorylation cycles. Mathematical modeling shows that the transition between high KaiA activity and low KaiA activity must depend on the total phosphorylation level of the KaiC hexamer in a switch-like or ultrasensitive fashion[36–40]. However, despite the importance of such a switch-like behavior for coherent oscillation in the KaiC hexamer, its molecular origin remains unclear. Since the phosphorylation of KaiC both stores information about timing of the circadian clock and determines the level of KaiA inhibition, the key questions are how phosphorylation alters the structure of KaiC, and how this structural change gives rise to the ultrasensitivity of KaiA activity switch.

These questions have been difficult to address. Previously reported high-resolution crystal structures with different phosphorylated and phosphomimetic states are nearly identical[14,15,26,41,42], indicating that there are functionally important conformational states of KaiC present in solution that are difficult to capture using crystallography. It has also been suggested that KaiC has dynamic structural properties that change across a phosphorylation cycle[29,43–45] with the flexibility of KaiC CII ring and structure of the A-loop directly affecting the KaiC phosphorylation activity[28,43]. However, the structure of dynamical states of KaiC and how KaiC monomers in a hexamer interact with each other are unclear. Here, we combine cryo-EM experiments, machine learning for data analysis, and theoretical modeling to address these two questions.

Cryo-EM is a powerful technique to resolve the structure of biological macromolecules. So far, most cryo-EM studies have focused on solving protein structure by averaging over a large number of projected single-particle images[46–54]. Here, we develop a new application of cryo-EM technology beyond structural analysis to understand subunit-subunit interactions in a protein complex by analysis and modeling the statistics of 3D structures in different conformational states. Our approach consists of two steps: (1) Collection of a large number of single-particle images by cryo-EM and the subsequent analysis of the large image dataset by using RELION[55,56] software to cluster the images into a small number of conformational states; (2) Modeling the distribution of hexameric particles in different conformational states by using a statistical physics model with a Hamiltonian that includes possible subunit-subunit interactions in the complex. Comparison between the distribution from the cryo-EM dataset and that from the model is used to determine the range and the strength of the subunit-subunit interactions.

In this paper, we report our investigations of statistical characteristics of the conformations of KaiC hexamer following the approach outlined above. To remove the complication of time-varying phosphorylation, we used two KaiC phosphorylation mutants in this study. One mimics the dawn-like dephosphorylated state via alanine substitution at the phosphorylation sites (KaiC-AA), the other mimics the dusk-like fully phosphorylated state via glutamate substitution (KaiC-EE). These mutants have been previously shown to differentially

allow KaiB binding[22,38], indicating that they capture the most salient features of KaiC at different times of day.

By analyzing a large number (millions) of cryo-EM images, we found that the structures of the A-loop in the KaiC monomers (millions) fall into two distinct states—some subunits have an A-loop that is largely buried in the subunit interface (buried state), and others have an extended, flexible A-loop protruding from the KaiC particle (exposed state). Both conformational states exist in each mutant with the exposed state favored in the KaiC-AA mutants and the buried state favored in the KaiC-EE mutant, which suggests that the two conformational states exist in a dynamic equilibrium modulated by the phosphorylation level of KaiC.

After identifying the conformational states of individual KaiC monomers, we investigate their spatial arrangements in all the KaiC hexamer rings. We find that there is substantial correlation in the ring with closer KaiC monomers (subunits) having a higher probability to adopt the same conformation. Based on this discovery, we propose an Ising model[57–59] to describe statistics of the conformational states found within each hexamer. We find that the observed frequency of hexamers with different patterns of conformational states can be described quantitatively by an Ising model with nearest neighbor interactions and a mutant-specific local field.

By combining statistical analysis of a large set of cryo-EM images and theoretical modeling, our study reveals that the A-loops within a KaiC hexamer form an intrinsically cooperative switch due to nearest neighbor subunit interactions, and the role of phosphorylation is to bias the switch towards the exposed or buried state.

## Results

### Cryo-EM structure determination of KaiC-AA and KaiC-EE

To investigate the effects of different phosphorylation states on structures and functional mechanisms in Kai system, two KaiC phosphomimetic mutants were used (KaiC-AA (S431A, T432A) and KaiC-EE (S431E, T432E)). These two mutations mimic the state of the clock near dawn and near dusk, respectively. We used a FEI Titan Krios G2 microscope device to collect cryo-EM data of KaiC-AA and KaiC-EE after incubation in the presence of 1 mM ATP (see Methods).

We first focused on two structures: one in KaiC-AA (Fig. 1a) and one in KaiC-EE (Fig. 1b), both refined to nominal resolution of 3.3 Å with relatively clear secondary structures (Fig. 1c, d, Supplementary Fig. 1). These two cryo-EM densities (Fig. 1a, b) were superimposed together for comparison (Supplementary Fig. 2a). We found that the KaiC-EE density is more compact in the CII domain. This observation is consistent with previous reported Trp fluorescence results showing that the overall shape of KaiC is more loosely packed in S/T state than in pS/pT state[44]. The difference map (Supplementary Fig. 2b) calculated by RELION[55,56] indicates that this density corresponds to a pair interacting loops, i.e., the A-loop (residue 488–497) and the 422-loop (residue 417–429). In the refined KaiC-EE atomic model, distance between G421 (belongs to the 422-loop) and S491 (belongs to the A-loop) is about 3.8 Å (Supplementary Fig. 2c).

### Two functional conformations for each KaiC subunit in a hexamer

To further study the dynamic behaviors of KaiC-AA and KaiC-EE, we focused on the CII domain using six-fold pseudo-symmetry (see Methods) by RELION[55,56]. There are two distinct possible conformations (exposed and buried) for each KaiC subunit in a KaiC hexamer (Fig. 2a). For the buried state, the A-loop forms a well-defined "U-shaped" line inside the central channel; this stable conformation contributes to a strong density in the electron density map. For the exposed state, the A-loop tends to stick out of the hexamer and is very dynamic. These flexible conformation states without strong density in the central channel are collectively referred to as the "exposed state".

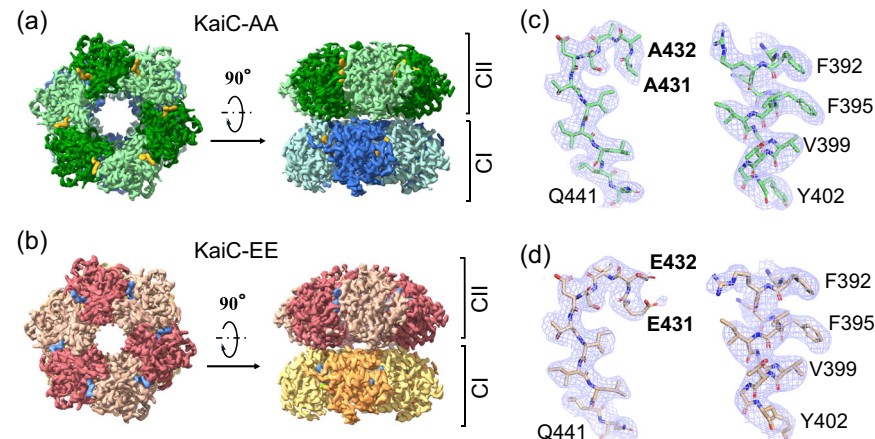

**Fig. 1 | Cryo-EM maps of KaiC-AA and KaiC-EE. a** Top and side view of KaiC-AA, with CII and CI domains colored in dark green and green and in blue and light blue, respectively. The 12 ATP molecules bound are colored in yellow. **b** Top and side view of KaiC-EE, with CII and CI domains colored in red and pink and in orange and yellow, respectively. The 12 ATP molecules bound are colored in blue. **c, d** show typical high-resolution densities of the secondary structures in the cryo-EM structure of KaiC-AA (**c**) and of KaiC-EE (**d**).

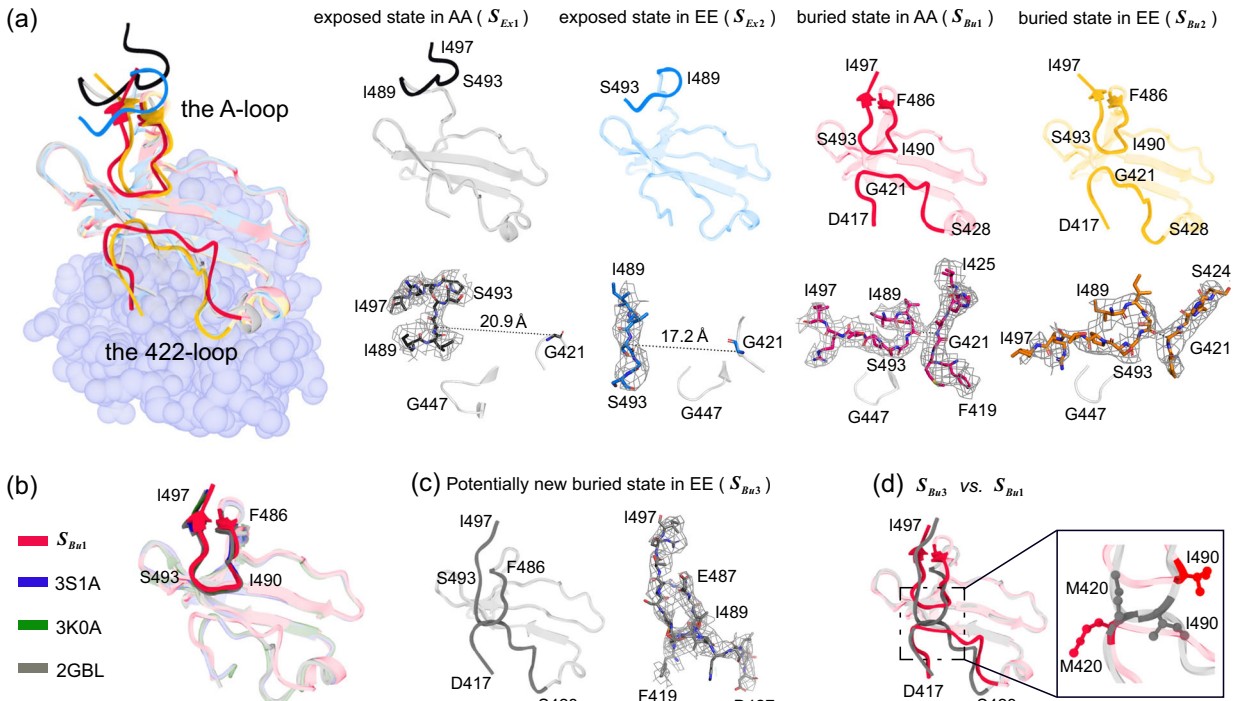

**Fig. 2 | Different conformational states of the A-loop and 422-loop area. a** (Left) The overview diagram of typical exposed and buried state in KaiC-AA and KaiC-EE. (Right) Corresponding independent exposed and buried state in KaiC-AA and KaiC-EE (denoted as $S_{Ex1}$, $S_{Ex2}$, $S_{Bu1}$, $S_{Bu2}$, colored in black, royal blue, red, orange, respectively). At the bottom are the close-up view of the A-loop shown in a stick representation superimposed over its corresponding cryo-EM densities in a gray mesh at 3σ, 5σ, 9σ, 9σ level representation, respectively. **b** The buried state ($S_{Bu1}$) superimposed with previously reported crystal structures in PDB data bank (with 3S1A, 3K0A, 2GBL colored in blue, green and gray, respectively). The A-loop is shown as cartoons without transparency. **c** A potentially new buried state conformation only found in KaiC-EE data set (denoted as $S_{Bu3}$, colored in dim gray), with the A-loop and 422-loop densities shown as a gray mesh at 11σ level on the right. **d** Superimposed the atomic model of $S_{Bu1}$ and $S_{Bu3}$, with side chains of M420 and I490 are given in a close-up view and labeled.

See Methods section and SI for quantitative criteria for distinguishing the buried and exposed states.

Given its exposed and dynamic structure, the A-loop in the exposed state is likely to have a stronger interaction with KaiA[24]. We also compared the structure of the buried state with previously reported crystal structures[14,15,26,41,42] and found that all structures were almost identical in the A-loop area (Fig. 2b, Supplementary Fig. 3). Thus, it seems reasonable that the crystal packing forces these crystal structures into what we call the buried state. However, it is worth noting that there is another type of buried state only observed in our KaiC-EE data set, i.e., a potentially new buried state in which the A-loop and 422-loop are directly connected (Fig. 2c). This is mainly due to movements of two residues: the Met420 side chain moves up while the Ile490 side chain moves down (Fig. 2d). This potentially new conformation indicates a stronger interaction between the A-loop and 422-loop that may be induced by Ser431 phosphorylation. This is consistent with the proposal that the A-loop restrains the motion of 422-loop thereby regulating the kinase reaction[60].

Consistent with our observation of conformational changes in the A-loops, KaiC-AA and KaiC-EE have been previously shown to bind KaiA with different affinities that are correlated with the sensitivity of the A-loop to proteolysis[61]. Another marked difference between KaiC-AA and KaiC-EE is their capacity to bind to KaiB[22,62]. This interaction requires ATP hydrolysis in the N-terminal CI domain[30], allowing KaiB to bind to a remodeled B-loop in that domain. In the absence of KaiB, we detect CI only in its pre-hydrolysis state for both KaiC-AA and KaiC-EE. This suggests that, in solution, the exposure of the B-loop is a transient event that can then be stabilized by KaiB binding[63].

## Strong conformational cooperativity in KaiC hexamers

By using RELION[55,56] to classify the original cryo-EM hexameric particles (1,592,573 KaiC-AA hexameric particles and 934,373 KaiC-EE hexameric particles), a subset of hexameric particles with balanced orientations were selected for 3D reconstruction. The criterion for particle selection is that the numbers of particles in different projection directions (top-view, tilt-view, side-view) need to be balanced in order to reconstruct the three-dimensional (3D) structures accurately[56]. In our experiments, the top-view particles are much more abundant than the non-top-view (side-view and tilt view) particles, which is commonly observed in KaiC cryo-EM studies[63]. The reason for this phenomenon is not clear at present, but we suspect that it may relate to the charge or hydrophilic/hydrophobic properties of protein surface (for example, top view is more hydrophobic and easier exposed to the gas-liquid interface), but this phenomenon should not strongly correlate with the conformational states of the protein. In our study, to balance the particle numbers in different orientations, we kept all the non-top-view particles (78,524 for KaiC-AA and 100,803 for KaiC-EE), and selected only a subset of top-view particles of comparable number as that of the non-top-view particles from the large pool of top-view particles. As we will show later in the paper, the specific choice of the subset of top-view particles does not change the statistics of the KaiC fine structures.

All the KaiC monomers in these selected hexameric particles are then refined and clustered into a number of clusters (16 used here), each of which is represented by the averaged structure (3D volume) of the individual hexameric particles belonging to the cluster. (See Supplementary Figs. 4 and 6 for the data processing flowcharts of KaiC-AA and KaiC-EE, respectively.) Next, each KaiC monomer cluster is classified as the buried (Bu) or exposed (Ex) conformational state based on the overlap between its 3D volume and two structure masks that characterize the Bu conformation (see Supplementary Figs. 5 and 7 for details). As a result of this analysis, we can assign each monomer in all the selected hexamers one of two states: Bu or Ex. There is a fraction of monomers (25.7% for KaiC-AA and 18.2% for KaiC-EE) that can not be classified as either Bu or Ex with sufficient statistical confidence. We call their conformation Undefined (Un). These Un monomers do not have a well-defined structure. They may represent the transitional state(s) between the Bu state and the Ex state or they could be caused by inaccuracy in our experiments.

We first studied the statistics of the conformational states of the KaiC monomers. We found that the probabilities of KaiC monomers being in the exposed or buried state depends on its phosphorylation state. KaiC-AA is more likely to be in the exposed state, whereas KaiC-EE is more likely to be in the buried state (see Supplementary Figs. 4 and 6 for details). Next, we investigated the statistics of the conformational states of the 6 subunits (monomers) in a hexamer. For a KaiC hexamer, there should be $2^6 = 64$ possible configurations (arrangements) of the 6 monomers, each with two possible conformational states. Considering the rotational degeneracy, these configurations can be combined into 13 conformational patterns each with a degeneracy index $\Omega_k (k = 1, 2, \ldots, 13)$, which corresponds to the number of configurations that pattern-k contains: $\sum_{k=1}^{13} \Omega_k = 64$ (see Sec. S2 in Supplementary Material for details). We put the conformational state of the monomers

back into their positions in hexamers and counted the probabilities of these 13 conformational patterns for KaiC-AA (Fig. 3a and KaiC-EE (Fig. 3b), respectively, for those hexamers in which all 6 monomers have clearly defined conformational states (24,240 hexameric particles for KaiC-AA, 116,785 hexameric particles for KaiC-EE). An interesting observation is that, if we assume the undefined monomers randomly occur in hexamers, the estimated numbers of clearly defined hexamers are $140475 \times (100\% - 25.7\%)^6 \approx 2.4 \times 10^4$ for KaiC-AA and $371557 \times (100\% - 18.2\%)^6 \approx 1.1 \times 10^5$ for KaiC-EE where 140475 and 371557 are the total number of particles (over all orientations) used for KaiC-AA and KaiC-EE analysis, respectively. This estimate agrees with the observed numbers, which lends additional support to the fact that the undefined (Un) monomer states are simply randomly unresolved structure and there are only two meaningful monomer conformational states (Bu and Ex).

Qualitatively, our analysis (Fig. 3a, b) shows that there is cooperativity in the conformational transitions of KaiC hexamer. Indeed, if all monomers in a hexamer were independent of each other, the probability distribution of the hexamer conformational patterns would follow a simple binary distribution: $P_k = \Omega_k p^{n_k}(1-p)^{6-n_k}$, where $p$ is the probability of a subunit in the exposed conformational state and $n_k$ is the number of exposed state in hexamer pattern-$k$ (see Supplementary Fig. 11b for values of $\Omega_k$ and $n_k$ for $k = 1, 2, \ldots, 13$). It is easy to see from our data that this is not the case. For example, the two hexamer conformational patterns 'Ex-Ex-Ex-Ex-Bu-Bu' ($k = 3$) and 'Ex-Ex-Ex-Bu-Ex-Bu' ($k = 4$) have the same $\Omega_3 = \Omega_4 = 6$ and $n_3 = n_4 = 4$, which would lead to their probabilities being equal without KaiC-KaiC cooperativity. However, our experiment results showed that $P_3 > P_4$ for both KaiC-AA and KaiC-EE ($\frac{P_3}{P_4} = 1.6(AA), 1.5(EE)$), which clearly indicates positive cooperativity among individual monomers that favors the neighboring monomers to be in the same conformational state, which may be related to the cooperativity in ATP hydrolysis[45].

To quantify this cooperativity, we calculated the pairwise (subunit-subunit) conformational correlation function for each dataset (EE or AA):

$$C(x) \equiv \langle S_{n,i} S_{n,i+x} \rangle_{n,i} - \langle S \rangle^2, \tag{1}$$

where $i (= 1, 2, \ldots, 6)$ is the ordered subunit index with periodic boundary condition ($S_{i+6} = S_i$) and $n$ represents different hexamers in the dataset. The state variable $S_{n,i}$ of subunit-i in the hexamer-n can be "+1" or "−1", corresponding to an exposed state or a buried state, respectively. $\langle S \rangle$ is the average state variable over all monomers in either Bu or Ex states, and the average $\langle S_{n,i} S_{n,i+x} \rangle_{n,i}$ is taken over all hexamers ($n$) and all pairs of monomers ($i, i + x$) in a hexamer except for those in which at least one of the monomers in the pair has an undefined conformation (Un). As is shown in Fig. 3c, the normalized correlation function $\tilde{C}(x) \equiv C(x)/C(0)$ is significantly larger than zero for $x \geq 1$, which clearly indicates subunit-subunit cooperativity.

To create more accurate 3D reconstructions, we have selected a group of particles with balanced orientations for our analysis. To make sure that particle orientation does not introduce bias in KaiC hexamer fine structure statistics, we have studied the statistics of thirteen hexamer conformational patterns for the top-view particles and the non-top-view (side-view and tilt-view) particles separately. As shown in Supplementary Fig. 8, the statistics of thirteen hexamer conformational patterns remain roughly the same for the top view particles and the non-top-view particles, which confirm that the particle orientation does not show significant correlation with the fine structure of the KaiC hexamers. Furthermore, to verify that the selection of a specific subset of top-view particles in the 3D reconstruction does not affect the statistics of the hexamer conformational patterns, we have also analyzed the statistics of conformational patterns by including other randomly selected subsets of top-view hexamer particles that are not used in the current 3D reconstruction (see Sec. S2 in Supplementary

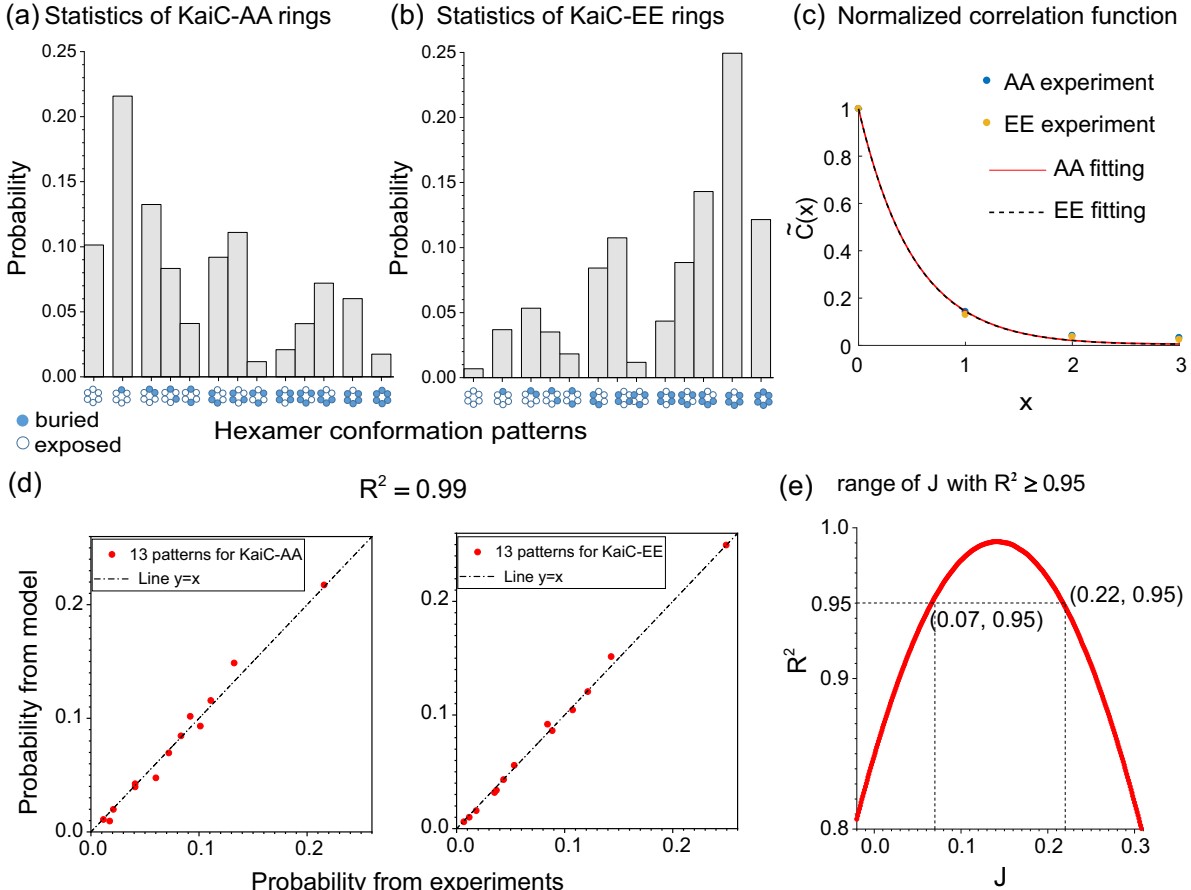

**Fig. 3 | Statistical results for pure KaiC-AA and KaiC-EE hexamers.** Probabilities of KaiC hexamers in different conformational patterns for KaiC-AA data set (**a**) (24,240 rings) and for KaiC-EE data set (**b**) (116,785 rings). The patterns are grouped and ordered according to the total number of exposed (or buried) states in the hexamer (total "spin"). Given the large number of hexamers in each pattern (>100 particles), the relative errors are small (<10%). **c** Normalized correlation function for KaiC-AA and KaiC-EE hexamers, respectively. Dots are calculated from

experimental data whereas lines are the theoretical results from the Ising model. About $10^6$ subunit pairs are used to compute $\widetilde{C}(x)$, so the error is ~$10^{-3}$, which is too small to show in the figure. **d** Comparison between experiment and model for KaiC-AA (left) and KaiC-EE (right), respectively. **e** Optimal $R^2$ varies with coupling constant $J$. In the range of $J \in [0.07, 0.22]$, $R^2$ is higher than 0.95, with the maximum at $J = 0.14$.

Information for details). The results indicated that including these other randomly selected top-view particles did not change the statistics (distribution) of the conformational states of the hexamers, i.e., the selection of particles for reconstruction purpose does not introduce a systematic bias for the conformational states of the hexamers.

## An Ising model quantitatively explains the experimental data

Next, we explain the observed statistics of the conformational states in KaiC hexamers (Fig. 3a–c) by using a simple model that includes cooperativity among monomers in a hexamer. Given the individual conformational state of each monomer in hexamers, the all-or-none Monod–Wyman–Changeux (MWC) model[64] certainly does not fit the experimental data. We thus adopted the more general Ising-type model[57–59,65], which is a minimal model that considers only the most salient features suggested by experiments: (1) the conformation of a subunit depends on its phosphorylation level; (2) there is a positive cooperativity between neighboring subunits in the hexamer.

In the minimal model, only the nearest neighbor subunits interact with each other, and the Hamiltonian (energy) of each configuration (64 in total) can be expressed as:

$$H(\vec{S}) = -J \sum_{<ij>} S_i S_j - B_{AA(EE)} \sum_{<i>} S_i, \qquad (2)$$

where $\Sigma_{\langle ij \rangle}$ represents a sum over all nearest neighboring pairs. $J$ is the coupling constant—a positive value of $J$ favors the two neighboring subunits to be in the same conformational state; $B_{AA}$ (or $B_{EE}$) is the "local field" for the KaiC-AA (or KaiC-EE) subunit—a positive (negative) local field favors the exposed (buried) state for the subunit. Assuming the system is at thermodynamic equilibrium, the probability of each hexamer conformational pattern can be expressed as:

$$p(k) = \Omega_k e^{-\beta H_k} / \sum_k \Omega_k e^{-\beta H_k}, \, k = 1, 2, \dots, 13, \qquad (3)$$

where $H_k$ is the energy for hexamer pattern-$k$, and $\beta = \frac{1}{k_B T}$ with $k_B$ the Boltzmann constant and $T$ an effective temperature. The effective thermal energy is set to be the energy unit ($k_B T = 1$) henceforth. Figure 3d demonstrate the best fit of the experimental data to our model. $R^2$ between the model prediction and the experimental data is 0.99 (Fig. 3d, consider both KaiC-AA and KaiC-EE together) indicating that the experimental data can be quantitatively described by a simple Ising model with only nearest neighbor interactions. In Fig. 3e, we plot the dependence of $R^2$ on the coupling constant $J$, which clearly shows that our data cannot be explained without cooperativity ($R^2 = 0.84$ when $J = 0$). The best parameters for fitting both the KaiC-AA and KaiC-EE mutant data are: $J = 0.14 \pm 0.07$, $B_{AA} = 0.19 \pm 0.04$, $B_{EE} = -0.25 \pm 0.04$ (error bars computed with $R^2 \geq 0.95$, see Fig. 3e). These parameters

indicate that there is substantial cooperativity between nearest neighbor subunits, and different local fields caused by Alanine or Glutamate mutation have opposite effects on the propensity of the exposed state or buried state.

The normalized correlation function can be determined exactly in the Ising model:

$$\widetilde{C}_{AA(EE)}(x) = \frac{\exp\left(-\frac{x}{\xi_{AA(EE)}}\right) + \exp\left(-\frac{6-x}{\xi_{AA(EE)}}\right)}{1 + \exp\left(-\frac{6}{\xi_{AA(EE)}}\right)},$$

where the correlation length for KaiC-AA and KaiC-EE are respectively: $\xi_{AA(EE)}^{-1} = \ln(\lambda_{AA(EE)}^{+}/\lambda_{AA(EE)}^{-})$, with $\lambda_{AA(EE)}^{\pm} = e^J \cosh B_{AA(EE)} \pm \left(e^{2J}\sinh^2 B_{AA(EE)} + e^{-2J}\right)^{\frac{1}{2}}$. We found that the correlation function from the Ising model fits the observed correlation accurately (Fig. 3c). The best fit parameters, $J = 0.15, B_{AA} = 0.21$, and $B_{EE} = -0.22$, are in quantitatively agreement with those obtained from fitting the configuration data (Fig. 3a, b, d). The correlation length is found to be short: $\xi_{AA} \approx \xi_{EE} \approx 0.5 < 1$, which confirms that the dominant subunit-subunit interactions are those between neighboring subunits in the KaiC hexamer.

Notice that the coupling constant $J$ is the same in KaiC-AA and KaiC-EE, which means that the Ala or Glu mutation only affects the propensity of each individual subunit, rather than subunit-subunit cooperativity. In a previous study[38], the subunit free energy difference between two functional states (competent and incompetent to interact with KaiB) is estimated to be $1 \pm 0.14$ for unphosphorylated (ST) KaiC and $-1 \pm 0.69$ for doubly phosphorylated (pSpT) KaiC based on modeling the observed oscillatory dynamics of the KaiC system. From the Ising model studied here, the free energy difference between the exposed state and the buried state for a single subunit is $\Delta H_{AA} = 2B_{AA} = 0.38 \pm 0.08$ and $\Delta H_{EE} = 2B_{EE} = -0.5 \pm 0.08$ for the AA (mimicking ST) and EE (mimicking pSpT) mutants, which are of the same order of magnitude as the previous estimates and their signs and relative values are consistent with the previous estimates. The quantitative difference may be due to differences between mutant (AA and EE) and wild-type (ST and pSpT) proteins.

During the circadian oscillation, KaiC hexamers likely contain mixtures of subunits with different phosphorylation levels. While the pure (homohexameric) phosphosite mutants clearly show cooperativity in our experiments, these mutants are extremes, in the sense that the non-mutant system is never fully phosphorylated. Thus, we next ask the question whether cooperativity between neighboring A-loops still exists in mixed KaiC hexamers where the monomers have heterogeneous phosphorylation levels.

To test the generality of cooperative interactions between neighboring KaiC monomers in the hexamer, we constructed a mixed sample with both KaiC-AA and KaiC-EE monomers[21] and measured the distribution of KaiC conformational states in hexamers. Briefly, KaiC mutants (KaiC-AA, KaiC-EE, 1:1) were buffer exchanged into the running buffer with 0.5 mM ADP, and incubated at 4 °C for about 24 h to disrupt hexamer structure. Then monomerized KaiC-AA and KaiC-EE were mixed before re-hexamerization via the addition of ATP[38] (see Methods). We collected cryo-EM data of this mixed sample with FEI Titan Krios G2 microscope. After unsupervised 2D classification and 3D refinement by RELION[55,56], we obtained the cryo-EM density map that was refined to nominal resolution of 3.8 Å (see Supplementary Fig. 9 for details). By following the same procedure for the pure samples, we obtained the probabilities of the 13 hexamer conformational patterns for the mixed sample, as shown in Fig. 4a.

We first examined the normalized correlation function $\widetilde{C}_{mix}(x)$, as shown in Fig. 4b. It's clear that $\widetilde{C}_{mix}(x)$ is non-zero for $x \geq 1$, but the correlation is weaker than that in the pure hexamer cases. With the

same fitting procedure for the correlation function as in the pure hexamer case, we obtained an effective coupling constant $J_{mix} = 0.086$, which is about a half of that for pure hexamers. Thus, the statistically significant correlation confirms the existence of subunit-subunit cooperativity in mixed hexamers. However, the strength of the overall cooperativity in the mixed hexamer as characterized by $J_{mix}$ is weaker.

To understand the detailed conformational pattern statistics of the mixed hexamers shown in Fig. 4a, we need to extend the Ising model for describing the mixed hexamers. In particular, besides its conformational state, a given monomer-$i$ in a mixed hexamer is characterized by its modification state (phosphorylation level) $\sigma_i$: $\sigma_i = 0 \, or \, 1$ if subunit-$i$ is KaiC-AA or KaiC-EE, respectively. As a result, each hexamer can have 14 possible spatial arrangements of monomer phosphorylation level (see Sec. S3 and Supplementary Table 2 for details), which is labeled by $l(l = 1, 2, \ldots, 14)$ with the probability of arrangement-$l$ denoted as $q_l$. For a given phosphorylation arrangement-$l$, the Hamiltonian of different hexamer conformational pattern $k$ ($k = 1, 2, \ldots, 13$) can be determined by the extended Ising model:

$$H_l\left(\vec{S}\right) = -J\sum_{<ij>} S_i S_j - \sum_i [B_{AA}(1 - \sigma_{i,l}) + B_{EE}\sigma_{i,l}]S_i, \quad (4)$$

where $\sigma_{i,l}(i = 1, 2, \ldots, 6)$ represents the modification state of the $i'th$ monomer in the hexamer ring in the phosphorylation arrangement-$l$. Note that the three parameters $(J, B_{AA}, B_{EE})$ are the same as those determined by using the previous experiments with pure KaiC-AA and KaiC-EE, which corresponds to one of the 14 modification state arrangements. Similar to Eq. (3), the probability of each hexamer conformational pattern $p_l(k)$ can be determined as:

$$p_l(k) = \Omega_k e^{-H_{l,k}} / \sum_k \Omega_k e^{-H_{l,k}}, k = 1, 2, \ldots, 13, \quad (5)$$

where $H_{l,k}$ is the energy for hexamer pattern-$k$ with subunit arrangement-$l$. The overall distribution of the hexamer conformational patterns is obtained by the weighted average:

$$P_k = \sum_l q_l p_l(k), \quad (6)$$

which depends on the distribution $q_l$.

Because we do not measure the detailed pattern of how differentially phosphorylated monomers are assembled into hexamers, we start with two extreme scenarios for mixing: one is the fully-mixed scenario, i.e., each monomer in the ring has an equal probability of being AA or EE; another is the no-mixing scenario, i.e., only pure hexamers (all EE or all AA) exist. We found that neither of these extreme scenarios agrees with the data: $R^2$ between the mixed experimental data and model prediction is 0.68 for no-mixing scenario (Fig. 4c) and 0.79 for fully-mixed scenario (Fig. 4d), see Supplementary Fig. 13 for the predicted distributions based on these scenarios. The relatively low accuracy in fitting the simple Ising model with a constant coupling constant suggested that interactions between subunits may depend on their phosphorylation state in addition to their A-loop conformations. We tested this hypothesis by modifying the coupling constant between EE subunit and AA subunit to take a different value $(J - \Delta J)$ from the coupling constant $(J)$ between the same types of subunits Under the fully-mixed scenario, we found that the best fit of this modified Ising model has a small positive $\Delta J = -0.09$, which is consistent with the decrease of the interaction strength in mixed hexamer, i.e., $J_{mix} < J$. However, the improvement in accuracy of fitting is small ($R^2$ is 0.83), which indicates that including a different interaction strength between AA and EE alone cannot explain the experimental data (see Sec. S3 in Supplementary Information for details).

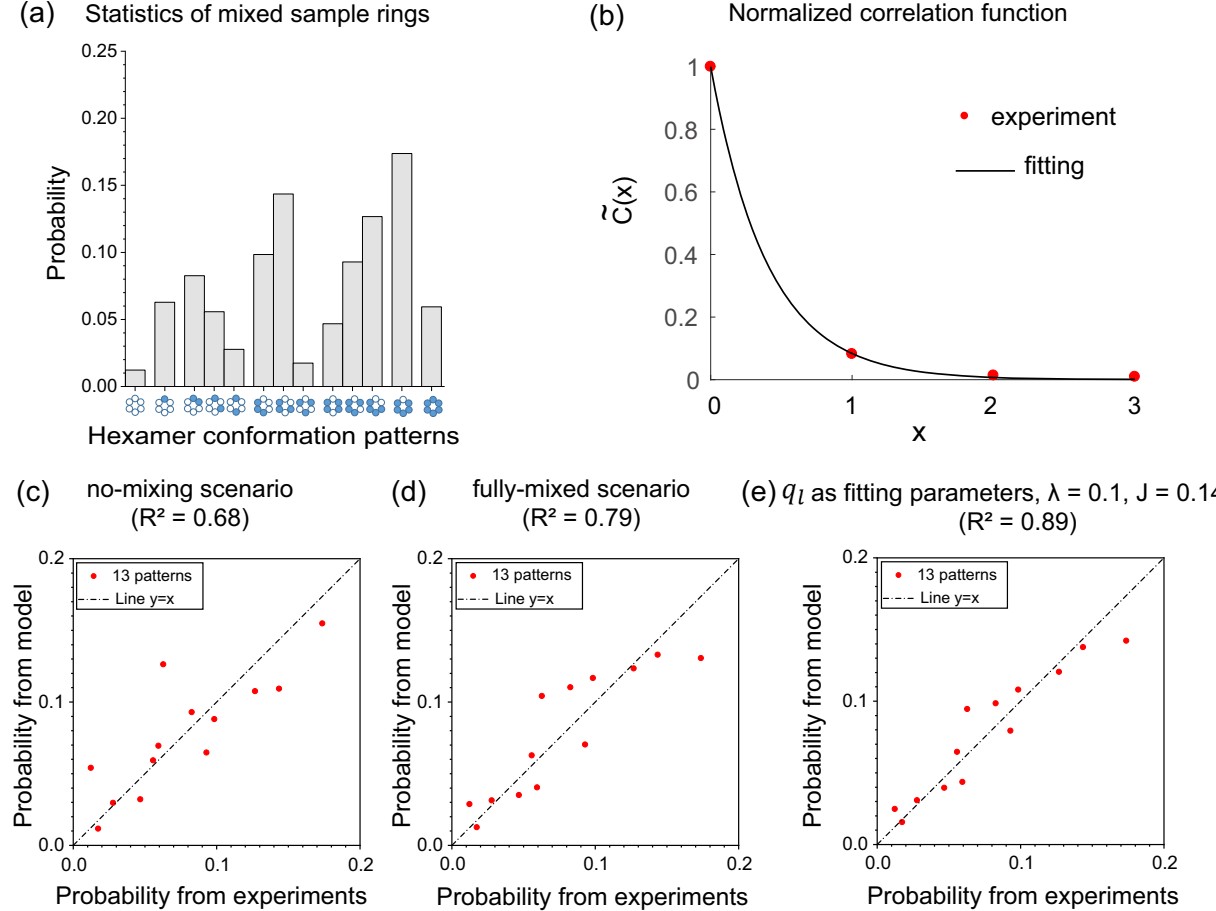

**Fig. 4 | Statistical results for the mixed hexamers. a** Probabilities of KaiC hexamers in different conformational patterns for the mixed sample data set (19,858 rings). Given the large number of hexamers in each pattern (>100 particles), the relative errors are small (<10%). **b** Normalized correlation function of the mixed sample. Dots are calculated from the experimental data while the solid line is from the fitting by the Ising model. About $10^6$ subunit pairs are used to calculate $\tilde{C}(x)$, so the error is $\sim10^{-3}$, too small to show in the figure. The fitting parameters are: $J_{mix} = 0.086, B_{mix} = -0.16$. Comparison between the mixed experimental data and model results with the no-mixing scenario (**c**), the fully-mixed scenario (**d**), and $q_l$ treated as free parameters with $\lambda = 0.1, J = 0.14$(**e**).

Our next strategy is to treat $q_l$ as fitting parameters. We obtained their values by fitting the theoretical predicted $P_k$ (from Eqs. 5 and 6) to experimental observation ($P_{k(ex)}$) subject to the constraints: $\sum_{l=1}^{14} q_l = 1, 0 \leq q_l \leq 1$; and the overall approximately equal percentage of KaiC-EE and KaiC-AA monomers over all hexamers (see Sec. S3 in Supplementary Material for details). As shown in Fig. 4e, the model results are in good agreement with the experiments: $R^2$ between the actual experimental data and model prediction data became 0.89. This agreement with experiments depends on the KaiC cooperativity. In the absence of the KaiC-KaiC interaction ($J = 0$), the agreement with experiments is poorer ($R^2 = 0.79$) even when we allow $q_l$ to vary (see Supplementary Fig. 14a for details). Furthermore, the resulting distribution $q_l$ for $J = 0$ is almost the same as the no-mixing extreme case with a very small fraction of mixed hexamers (see Supplementary Fig. 15a for details), which is clearly inconsistent with our results for pure hexamers that show strong cooperativity. The good fit to data with a finite cooperativity $J$ again confirms the existence of the monomer-monomer cooperativity, and this key result is robust when the weight constant ($\lambda$) in our optimization algorithm takes on different values (see Supplementary Figs. 14 and 15b for details).

## Discussion

In this work, we developed a general approach by combining cryo-EM imaging technology with machine-learning based image analysis and statistical physics-based modeling to unravel possible subunit-subunit interaction in a protein. We applied this general approach to study

cooperativity in KaiC hexamer, which is the key protein complex in the Cyanobacteria circadian clock.

We report two distinct functional conformational states (exposed and buried) for each KaiC monomer that can coexist in a KaiC hexamer. Statistical analysis of single particle data suggests that there is a dynamic equilibrium between the two conformational states with the highly phosphorylated (dephosphorylated) KaiC prefers the buried (exposed) states, respectively. Characterization of the spatial arrangements of the exposed and buried states along with theoretical modeling reveal that there is substantial cooperativity in the hexamer that favors the neighboring KaiC monomers to be in the same conformational state. The day-night switch in Kai system is crucial in maintaining circadian rhythms, related to a general requirement for nonlinear response in the feedback loops that support chemical oscillators[34,35,38,40,66]. The cooperative interactions between neighboring KaiC subunits as identified in this study can make the transition from buried state to exposed state sharper, described by a higher Hill coefficient (see Supplementary Fig. 16 for details), and thus provide a possible molecular mechanism for controlling the day-night switch. The picture that emerges from our analysis is that interactions between the C-terminal regions within the KaiC hexamer form an intrinsically cooperative switch. The role of changing phosphorylation is then to shift the midpoint of the switch, ultimately actuating KaiB binding to initiate the next cycle.

Recently, Jeffrey A. Swan et al.[63] studied the structure of KaiC-AE and KaiC-EA mutants, mimicking the state of the clock near noon and

near midnight, respectively. Their results also confirm that CII has two possible subunit conformations, whose distribution depends on the phosphorylation in the CII ring. In addition, change of phosphorylation in CII leads to a hexameric (global) conformational change, which finally affects KaiB affinity. Our results are consistent with their work. Furthermore, the cooperative subunit conformational switch found here provides a possible molecular mechanism for the switch in hexameric conformation that controls KaiB binding.

Our analysis of the mixed sample experiments suggests that the mixing of monomers with different phosphorylation levels may not be random. The enrichment factor α (the ratio of optimized $q_l$ values when $\lambda = 0.1$ (Supplementary Fig. 15b) to that under fully-mixed scenario (Supplementary Table 2)) indicates that there is a higher probability for two neighboring subunits to have different phosphorylation levels (Supplementary Fig. 15c). This possible preferential mixing phenomenon is worth further investigation given that KaiC hexamers constantly disassemble and reassemble during circadian oscillation through monomer shuffling[27,67,68], which is thought to be involved in synchronization[69–71]. In addition, we found that the effective coupling constant $J_{mix}$ in the Ising model for the mixed KaiC case is weaker than that in the pure hexamer case, which provide an alternative explanation for the reduced cooperativity due to a weaker coupling between heterogeneous KaiC monomers. Indeed, the microscopic origin of the nearest neighbor KaiC interaction that leads to cooperativity, which is key for coherent oscillations, is an important open question for future studies.

In general, the combination of cryo-EM technology for collecting a large number of single-particle images, together with machine learning based image analysis and statistical physics based modeling provides a powerful tool for deciphering interactions within functional protein complexes, which is otherwise hard to probe directly.

## Methods

### Protein expression and purification
KaiC-AA and KaiC-EE were expressed and purified as described previously[38]. Briefly, KaiC mutants were expressed recombinantly in *E. coli* with N-terminal His$_6$ tags. Clarified lysate was purified using Ni affinity chromatography (Histrap, Cytiva), fractions containing KaiC were pooled, and the expression tags were cleaved overnight. The resulting material was further purified using size exclusion chromatography (HiPrep S300, Cytiva) and fractions corresponding to the molecular weight of KaiC hexamers were selected.

### Cryo-EM imaging and data collection
To remove the glycerol, KaiC proteins were applied to Zeba Micro Spin Desalting Columns (7 K, Thermo Fisher), exchanging the buffer to running buffer (20 mM Tris-HCl (pH 8.0), 150 mM NaCl, 5 mM MgCl2, 0.5 mM EDTA, 1 mM ATP). Then KaiC-AA (or KaiC-EE) was diluted to 0.35 ug/ul with running buffer and incubated at 30 °C for 6 h to ensure that KaiC equilibrates to a functionally relevant state. Cryo-EM grids were prepared with FEI Vitrobot Mark IV. QUANTIFOIL grids (R2/1, 300 Mesh) were glow-discharged before a 3.5-µl drop of 0.35 ug/ul KaiC-AA (or KaiC-EE) solution was applied to the grids in an environmentally controlled chamber with 100% humidity and 4 °C temperature. After 1 blot force, 1 s blot time, the grid was plunged into liquid ethane and then was transferred to liquid nitrogen. The cryo-EM data was collected on a FEI Titan Krios G2 microscope connected to Gatan K2 Summit direct electron detector in a super-resolution counting mode, using SerialEM[72] semi-automatically. Coma-free alignment was manually optimized and parallel illumination was verified before data collection. A total exposure time of 10 s with 250 ms per frame resulted in a 40-frame movie per exposure with an accumulated dose of ~50 electrons/Å$^2$ (see Supplementary Table 1). The calibrated physical pixel size and the super-resolution pixel size are 1.37 Å and 0.685 Å,

respectively. Raw data were saved at the pixel size of 0.685 Å. A total of 5125 movies of KaiC-AA and 3530 movies of KaiC-EE were collected.

### Preparation of mixed hexamer sample
The mixed sample was made following the method of ref. 21 to obtain KaiC-AA and KaiC-EE monomers. Protein concentrations were quantified by BCA Protein Assay Kit to prepare a 1:1 molar ratio of KaiC-AA to KaiC-EE. KaiC-AA (KaiC-EE) was buffer exchanged twice with Zeba Micro Spin Desalting Columns (7 K, Thermo Fisher) into a buffer where ATP was replaced with 0.5 mM ADP, incubated at 4 °C for about 24 h to disrupt hexamer structures. Then monomerized KaiC-AA and KaiC-EE were mixed for re-hexamerization by buffer exchanging into reaction buffer with 5 mM ATP[38]. Total KaiC concentration was quantified after re-hexamerization by BCA Protein Assay Kit, then diluted to 0.35 ug/ul with running buffer and incubated at 30 °C for 6 h. Cryo-EM grid preparations and data collection procedures are the same as previously described.

### Cryo-EM data processing
All frames of raw movies were aligned and averaged with the MotionCor2 program[73] at a super-resolution pixel size of 0.685 Å. Each drift-corrected micrograph was used for the determination of the micrograph CTF parameters with program Gctf[74]. We picked 1,592,573 hexameric particles of the KaiC-AA, 934,373 hexameric particles of the KaiC-EE, 693,666 hexameric particles of the mixed hexamer sample using the program EMAN2[75]. Reference-free 2D classification and 3D classification were carried out with two-fold binned data with a pixel size of 1.37 Å in both RELION[55,56] and ROME[76]. Focused 3D classification, which we used in the later stage of data processing, and high-resolution refinement were mainly conducted with RELION[55,56]. A substantial part of the data processing, mostly 2D and 3D classifications, were performed with clusters supported by High Performance Computing Platform in PKU.

There were 140,475 hexameric particles of KaiC-AA, 181,326 hexameric particles of KaiC-EE, 175,284 hexameric particles of the mixed hexamer sample in the dataset chosen for the following steps of analysis. The final refinement was done using data with a pixel size of 1.37 Å that were binned by two-fold from the raw data in the super-counting mode. Based on the in-plane shift and Euler angle of each particle from the last iteration of refinement, we reconstructed the two half-maps of each structure using raw single particle images at the super-resolution mode with a pixel size of 0.685 Å, which resulted in reconstructions for the KaiC-AA, KaiC-EE, and the mixed sample with overall resolutions of 3.3 Å, 3.3 Å and 3.8 Å, respectively, measured by gold-standard FSC at 0.143-cutoff. All density maps were sharpened by applying a negative B-factor −100 manually. Local resolution variations were further estimated using ResMap[77].

### Atomic model building and refinement
To build the initial atomic models of KaiC-AA and KaiC-EE, we used a previously published KaiC structure[42] and then manually improved the main-chain and side-chain fitting in Coot[78] to generate the starting coordinate files. To fit the KaiC-AA and KaiC-EE atomic models to the corresponding reconstructed density maps, we first conducted rigid body fitting of the segments of the model in Chimera[79], after which the fitting of atomic models with density maps were improved manually in Coot. Atomic models of $S_{Ex1}$, $S_{Ex2}$, $S_{Bu1}$, $S_{Bu2}$, $S_{Bu3}$ were fitted in Coot[78] manually starting from the KaiC-EE structure. Finally, atomic models were all subjected to the real-space refinement program in Phenix[80], see Supplementary Table 1 for validation statistics.

### Structural analysis and visualization
Structural comparison was conducted in Pymol[81], Chimera[79], and ChimeraX[82]. All figures of the structures were plotted using Pymol[81], Chimera[79] and ChimeraX[82].

## Six-fold pseudo-symmetry and classification results by RELION

We used 140,475 hexameric particles of KaiC-AA (Supplementary Fig. 4), 371,557 hexameric particles of KaiC-EE (two remaining classes were added, which also resulted in good-quality refinement with overall resolutions of 3.8 Å) (Supplementary Fig. 6), and 175,284 hexameric particles of the mixed sample (Supplementary Fig. 9d) to perform C6 pseudo-symmetry expansions in RELION[55,56]. That is rotating each particle by 60°, 120°, 180°, 240°, 300° to get the symmetric copies and expanding the particle set. In this way, the whole data set was expanded 6 times. We used focused 3D classification (with CII domain masked) to further classify particles into two different conformational states (exposed state and buried state) using RELION[55,56], see Supplementary Figs. 5, 7, and 10 in Supplementary Material for the criterion of these states. We also tried C2 pseudo-symmetry and C3 pseudo-symmetry with both CI and CII domains, but didn't find significant amounts of particles with these lower symmetries.

## Criteria for distinguishing the exposed (Ex) state and buried (Bu) state

For each dataset (AA, EE, and mixed), we used the average 3D volumes of those clusters with the most well-established buried structure within the A-loop area (cluster 1-4 in Supplementary Fig. 5 for KaiC-AA; clusters 1–3, 1–7, 1–8, 1–10, 1–11 and 1–13 in Supplementary Fig. 7 for KaiC-EE; clusters 1–3, 1–6, 1–15 in Supplementary Fig. 10 for mixed KaiC) to create a black-and-white mask (digital map) by using a density threshold (0.025 for KaiC-AA, 0.2 for KaiC-EE, 0.1 for mixed KaiC) in RELION[55,56]. We then extend the white volume by one (or two) pixels in all directions to obtain mask1 (or mask2). The overlap intensities (or the integral density values) of the n-th 3D volume with mask1 (or mask2) is defined as $I_{1,n} = \sum_i D_{i,n}M_{1,i}$ (or $I_{2,n} = \sum_i D_{i,n}M_{2,i}$), $D_{i,n}$ is the density value of the n-th 3D volume at the i-th pixel point and $M_{1,i}$ ($M_{2,i}$) is a binary number that is 1 inside the white volume of mask 1 (mask 2) and 0 in the black volume of mask 1 (mask 2). The 3D volumes with larger integral density values of $I_{1,n}$ and $I_{2,n}$ are more likely to be in the buried (Bu) state, and those with smaller values of $I_{1,n}$ and $I_{2,n}$ are more likely to be in the extended (Ex) state. To increase the statistical confidence of the classification of the two states (Ex and Bu), we consider the 3D volumes with intermediate values of $I_{1,n}$ and $I_{2,n}$ to have an undefined (Un) conformational state. See Supplementary Figs. 5a, 7a, and 10a for details. To check the robustness and consistency of the classification of the buried (Bu) and the exposed (Ex) state, and more importantly, to independently judge the state of those 3D volumes at the boundaries between Un and either the Bu or the Ex states, each 3D volume is shown (only within the region corresponding to mask2) at high density threshold ($4\sigma$) and low density threshold ($2\sigma$), where $\sigma$ is the standard deviation of the 3D volume (density) within the hexamer region, see Supplementary Figs. 5b, 7b, and 10b for details. These 3D volumes are inspected to help determine their conformational states, especially at the boundary between the Bu state and the Un state where the determination is difficult by using the overlap intensities alone.

## Reporting summary

Further information on research design is available in the Nature Portfolio Reporting Summary linked to this article.

## Data availability

The data that support this study are available from the corresponding authors upon request. The cryo-EM maps have been deposited in the Electron Microscopy Data Bank (EMDB) under accession codes EMD-32952 (KaiC-AA) and EMD-32953 (KaiC-EE). The corresponding atomic coordinates have been deposited in the Protein Data Bank (PDB) under accession codes 7X1Y (KaiC-AA) and 7X1Z (KaiC-EE). Source data are provided with this paper.

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

## Acknowledgements
The authors thank Y.D.Mao, Y.N.Zhu, B.Liu, S.W.Zhang, S.T. Zou, H.Liang, Y.H.Wang, K.Y.Wang, D.Y.Yin, and W.L.Wang for constructive discussions; Y.Ma, X.Li, and X.Pei for technical supports. The cryo-EM data were collected from the Electron Microscopy Laboratory and cryo-EM Platform at Peking University. Data processing was performed on the Weiming No.1 and Life Science No. 1 High Performance Computing Platform in Peking University. The work by D.Z. is supported by the China Postdoctoral Science Foundation (2020M680180). M.J.Rust is supported by an HHMI-Simons Faculty Scholar award and NIH R01 GM107369. The work by Y.T. is supported by a NIH grant (R35GM131734). The work by Q.O. is supported by the National Natural Science Foundation of China (12090054).

## Author contributions
L.H. purified proteins; X.H. prepared samples for imaging and collected data; X.H. and D.Y. processed data and refined the maps; X.H. and D.Z. did the simulations and analyzed the data; Z.W. and T.Y. contributed to simulations and data analysis; M.R., Y.T., and Q.O. initiated the project, developed the model and analyzed the data; all wrote the paper.

## Competing interests
The authors declare no competing interests.
