## [Peer Review File · Nature Communications]

Determining protein-protein interaction from statistics of cryo-EM images: Observation of nearest-neighbor coupling in a circadian clock protein complexReviewers' Comments:

Reviewer #1:

Remarks to the Author:

Dear Editor,

In the manuscript "Determining protein-protein interaction from statistics of cryo-EM images: Observation of nearest-neighbor coupling in a circadian clock protein complex", Han et. al. reported a new approach to determine protein-protein interaction in a protein complex by combining cryo-EM imaging technology, machine learning, and statistical modeling. The authors used the cryo-EM single particle average method studied the circadian clock protein KaiC via collection of millions of KaiC monomer images. By selecting a small portion of particles with structural homogeneity, the subpopulation of the particles was classified into two major conformational states of KaiC monomer subunits based on the A-loop (I490-I495) conformations, which displayed as thirteen hexamer conformational states. Authors further used the thermal dynamics model analyzed the cooperativity between neighboring subunits and proposed a switch-like manner of phosphorylation process. Although the large cryo-EM data and statistics have been conducted, the manuscript is well written, few critical points showed below should be addressed before supported.

Major comments:

1. The structures of KaiC (including binding to KaiB) have well studied by crystal structure. The only new of this study is the conformational change of A-loop (indicated by mutation instead of binding protein) and the related analysis based on physics model under thermal dynamic equilibrium condition. Considering, only ~10% up to 40% of particles with structural homogeneity were selected for cryo-EM 3D reconstruction based on the overall shape, the selected particles seriously questioned to be used to representing the thermal dynamics of the particles in solution.
2. As authors described that the physics model and energy, such as Lsing Model and Hamiltonian are established under the thermal dynamic equilibrium condition (as descript in page 11, line 5). Authors should evidence that the classification and selection process in the cryo-EM reconstruction from a structural homogenous pollution would NOT affect the distribution of the A-loop conformation in solution. Without this classification selection is an unbiased sampling, all related analyses by statistic physics model is improperly.
3. The resolution/map used define the criteria of Bu/Un/Ex conformation is unconvincing, for example, Un 1-7 is similar to Bu 1-12 showed in Supp Fig. 9., It is knowing the resolution of the central portion of particle (such as A-loop) is lower than that near the edge. How the resolution variety affects the criteria of A-loop conformations.
4. Authors state "Previously reported high-resolution crystal structures with different phosphorylated and phosphomimetic states are nearly identical^{14,15,26,41,42}, indicating that there are functionally important conformational states of KaiC present in solution that are difficult to capture using crystallography". Author ignored the important crystal structure here (ref; 35, PDB: 3JWQ), in which the crystal structure did showed the conformal change of KaiC by directly binding to KaiB. 3JWQ showed the major conformation induced by binding KaiB is in the loop (Q116-D123) of CI instead of A-loop of CII discovered by cryo-EM average on the mutation sample with statistics. Authors should careful discuss why the conformational change on the CI domain has not been discovered from their mutation sample and selected particles.
5. The directly evidence from the KaiC binding to KaiA and KaiB should be studied to validate the conformational change revealed from the mutation samples.
6. Abstract stated the "uncover protein-protein interactions within multi-component biological nanomachines", there is no protein (such as KaiA or KaiB) interaction to the hexamer conformation of

KaiC.

7. Author stated, "an unsupervised machine learning method" has been used, however, no detailed information about this method, reference, or validation has been described in the manuscript.

Reviewer #3:

Remarks to the Author:

The authors employed cryo-EM and statistical modeling to analyze protein-protein interactions within hexameric KaiC, a protein complex central to maintaining the circadian rhythm in cyanobacteria. The authors identified two main conformation states of KaiC monomers, exposed and buried, and found strong evidence for cooperativity between the conformational states of neighboring monomers within hexamers. The experimental data fits beautifully with the Ising model the authors define in this work, showing that the observed conformational states can be confidently explained by nearest-neighbor subunit cooperativity. Overall, this is a highly original study, and a paradigm for the use of cryo-EM to quantitatively infer protein-protein interactions. That said, the clarity of presentation of some points could be improved, and the analysis of the mixed samples (i.e. composed of monomers of different phosphomimetic mutants) could be made stronger.

Major issues

1. Overall consistency of illustrating "well-resolved" structures:

Given the strategy of the authors' experiments (creating two mutants to mimic two phosphorylation states of KaiC) visual comparative analysis is critical. Figure 1, panels C and D show the same selected secondary structures for, respectively, the KaiC-AA and KaiC-EE phosphomimetic mutants, and this presentation is sufficient for visual comparison. However, panel D also includes additional secondary structures of the A-loop and 422-loop for EE which were not shown for AA. Given the limitation of obtaining the structure of highly dynamic or disordered protein regions, it is not surprising that the more exposed KaiC-AA A-loop was not well-defined in the experimental data. The lack of structure images for the A-loop and 422-loop in KaiC-AA (Figure 1) is seemingly justified in the following paragraph with this argument. However, Figure 2 then appears to show A-loop structures for KaiC-AA. This is inconsistent and confusing. I would expect that either these structures are not well-defined and should not be shown at all, or they are resolved well enough to be shown consistently throughout the manuscript. This point requires clarification.

2. Analysis of the mixed sample experiments.

In addition to analyzing hexamers composed exclusively of KaiC-AA or KaiC-EE, the authors also study mixed 50:50 samples of these two mutants. They first measure the correlation function for exposed and buried conformations and fit this to an Ising model, finding a coupling J that is lower than for the pure samples. I would have expected the authors to have then attempted to fit their data by allowing for a reduced J between nearest neighbors of different phosphomimetic states. Indeed, the authors write "...interactions between subunits may depend on their phosphorylation state in addition to their A-loop conformations" and refer to Supplementary Section 3 for incorporation of this dependence. However, this analysis does not appear anywhere in the supplementary information. As it is plausible that the value of J for unlike neighbors will differ from that for like neighbors, this analysis should be performed. Specifically, the authors should report the best fit to the cryo data assuming random arrangements of AA and EE monomers within hexamers, but allowing for a different J between AA and EE monomers, compared to the value of $J = 0.15$ between like monomers found in the pure samples. Even if this approach fails to produce a good fit to the data, the results should be reported, since even a negative result would be informative about how subunits are assorted within hexamers. A related question: Why don't the authors know the arrangement of AA and EE monomers within each

hexamer? I would have thought this information might be accessible from the cryo images. Such information would of course greatly inform the model for the mixed sample experiments. I can only assume that this information is not available for some reason, otherwise the authors would have used this important information in their analysis. This point needs to be clarified.

Minor issues

3. Confusing figure - suggested edits for overall clarity (Fig. 1):

- In panels A and B, the hexameric structures of KaiC-AA and KaiC-EE are shown in green/blue and red/yellow respectively. However, within the KaiC-AA structure (panel A) there are small yellow regions between individual monomeric units. Similarly, in panel B there are small blue regions between individual monomeric units. These small colored regions were not defined in the figure caption. Maybe they are the AA and EE residues? Needs clarification.
- It is confusing that Fig. 1 panel D has the A-loop and 422-loop structures alongside the KaiC-EE secondary structures while panel C does not. From my understanding, this is because the A-loop of the KaiC-AA mutant is dynamic and therefore unresolved.
 - o At a minimum, separating the A-loop and 422-loop structures into a distinct panel (i.e., panel E) would help with overall clarity.
 - o The secondary structures of KaiC-AA and KaiC-EE which show the mutation sites (panels C and D left) could be improved by (1) showing in panels A and B where in the hexamer these structures occur (i.e., inside the ring, outside the ring, between monomeric subunits, etc.) and (2) in C and D bolding the A432/A431 and E432/E431 text labels to distinguish them from the other labeled residues which are not mentioned in the main text.

4. Additional statistical calculation to bolster argument (second paragraph, page 9):

- The authors focused their analysis on those hexamers for which all monomers have clearly defined conformational states (24,240 particles for KaiC-AA and 116,785 particles for KaiC-EE). I did a quick calculation starting from the total number of hexamers in each experiment (~1,590,000 for AA and 934,000 for EE) and assuming the occurrence of undefined monomeric conformational states is random and not due to the existence of additional transition states. This yielded that the expected numbers of fully defined hexamers would be ~23,000 for KaiC-AA and ~113,000 for KaiC-EE given the starting number of particles. Given the close agreement of this random expectation with the observed numbers, the natural conclusion is that undefined states are simply randomly unresolved, rather than reflected additional conformational states that are unaccounted for. I suggest the authors redo my calculation more carefully, and if they agree, report this result as supportive of the conclusion that there are only two meaningful monomer conformations.

5. Additional analysis (below equation 1, page 10):

The conformational correlation function was described accounting for the hexamers where at least one of the monomeric subunits had an undefined conformation. What does the correlation function look like only using hexamers with no undefined monomeric subunits?

6. Weak comparison statement: (page 12, last paragraph):

- Authors cite Lin et al. (2014) as estimating the subunit free energy difference between the two functional states (buried and exposed) to be 1 ± 0.14 for ST and -1 ± 0.69 for pSpT. From the Ising model the authors describe, they calculate the free energy differences to be 0.38 ± 0.08 for ST mimic (KaiC-AA) and -0.5 ± 0.08 for pSpT mimic (KaiC-EE). Authors state that these values are "consistent with the previous estimates". While these numbers may have the same sign and relative magnitude, it is somewhat overstated to say they are "consistent". The authors then detail the "quantitative difference" in the following sentence which contradicts the previous claim that the values

are "consistent". The claim of consistency here should be restated more accurately.

7. Terminology/wording: (first paragraph, page 9):

- Authors state: "There is a 'small fraction' of monomers (25.7% for KaiC-AA and 18.2% for KaiC-EE)..." . A fraction of 26% is not so "small". Restate more accurately.

8. Terminology/wording: (bottom of page 8):

- The term "particle" here refers to the hexameric structure seen on the cryo-EM grids but is not explicitly defined and can therefore be misinterpreted as referring to a monomeric subunit. Without the definition of "particles" equating to the hexamer, it appears highly inconsistent when authors say "...a subset of particles with high quality well-defined structures" were selected and later state "...Un monomers do not have a well-defined structure". I suggest the authors consistently use the terminology "hexameric particles".

9. Terminology/wording (second paragraph of page 4):

- Authors state "...whether KaiC monomers in a hexamer interact with each other are unclear". The term "interact" is confusing in this sentence. By definition, individual subunits of a multimer must interact with each other.

10. Confusing figure panel- suggested edit for overall clarity (Fig 2)

- Panel B might be better presented as one structure superposition (rather than three individual ones). Given that SBU1 is identical in each overlay, one simple superposition with the same color legend for each of the three PDB structures would suffice.

11. Confusing figure element- suggested edit for clarity (Fig 3)

- Remove "(pure sample)" from panel C which is confusing and extraneous when not directly comparing this panel to panel B of Figure 4.

12. Confusing figure element - suggested edit for clarity (Fig 4)

- Remove "(mixed sample)" from panel B which is confusing when not directly comparing this panel to panel C of Figure 3.

13. Typo (bottom of page 3):

- "physophorylation" to "phosphorylation"

Reviewer #4:

Remarks to the Author:

The manuscript by Han et al makes important progress towards understanding how the protein nanomachine that comprises the cyanobacterial circadian oscillator is able to keep 24-h time and maintain its observed coherence in the absence of external synchronizing inputs. A wealth of biochemical and structural data over the past 20 years has established the fundamental structures and phosphorylation states that the KaiC hexamer navigates during the cycle, as it complexes with its partners KaiA and KaiB. However, missing from the existing catalog is a measure of the interactions among monomers within KaiC hexamers, and how each is affected by, and/or affects, the phosphorylation state of its neighbors. It has been long recognized that hexamers are mixed with

respect to phosphorylation state, but the current data have not addressed either the measurement or implications of such mixtures. The current manuscript addresses the problem by use of Cryo-EM to model the conformations of individual KaiC particles, avoiding the structural constraints that can be imposed by crystallization. The authors use two phosphomimetics that mimic "dawn" and "dusk" states of KaiC and derive key structural features, an exposed and buried state that includes the known A-loop, for each pure hexamer. By classifying mixed hexamers, they are able to conclude that there is significant cooperativity among monomers, which favors neighboring KaiC monomers to be in the same conformational state. The work helps to explain how KaiC mixtures retain coherence even though hexamers assemble and disassemble during the cycle. The work also suggests a mechanism for sharpening the "day-night switch" when KaiC transitions from its phosphorylation to dephosphorylation mode. The work is consistent with prior studies that, in more qualitative or limited ways, predicted the results that were obtained here. This project will inform models that can better explain the mechanism of the clock, whose dynamic structures, rife with allostery and cooperativity, hold the answers to how it tells time. The primary limitation (true of almost all studies on this protein) is that it depends on the phosphomimetic variants of KaiC, which imperfectly reflect KaiC dynamics. However, the current work is a necessary step on the way to understanding how the KaiA, KaiB, KaiC multi-protein mixture changes over time. Overall, this is a very important contribution to understanding the mechanism of the clock.

Response to reviewer #1's comments

(reviewer comments – black; response – blue; revisions made – red)

Reviewer #1 (Remarks to the Author):

In the manuscript “Determining protein-protein interaction from statistics of cryo-EM images: Observation of nearest-neighbor coupling in a circadian clock protein complex”, Han et. al. reported a new approach to determine protein-protein interaction in a protein complex by combining cryo-EM imaging technology, machine learning, and statistical modeling. The authors used the cryo-EM single particle average method studied the circadian clock protein KaiC via collection of millions of KaiC monomer images. By selecting a small portion of particles with structural homogeneity, the subpopulation of the particles was classified into two major conformational states of KaiC monomer subunits based on the A-loop (I490-I495) conformations, which displayed as thirteen hexamer conformational states. Authors further used the thermal dynamics model analyzed the cooperativity between neighboring subunits and proposed a switch-like manner of phosphorylation process. Although the large cryo-EM data and statistics have been conducted, the manuscript is well written, few critical points showed below should be addressed before supported.

Major comments:

1. The structures of KaiC (including binding to KaiB) have well studied by crystal structure. The only new of this study is the conformational change of A-loop (indicated by mutation instead of binding protein) and the related analysis based on physics model under thermal dynamic equilibrium condition. Considering, only ~10% up to 40% of particles with structural homogeneity were selected for cryo-EM 3D reconstruction based on the overall shape, the selected particles seriously questioned to be used to representing the thermal dynamics of the particles in solution.

2. As authors described that the physics model and energy, such as Lsing Model and Hamiltonian are established under the thermal dynamic equilibrium condition (as descript in page 11, line 5). Authors should evidence that the classification and selection process in the cryo-EM reconstruction from a structural homogenous pollution would NOT affect the distribution of the A-loop conformation in solution. Without this classification selection is an unbiased sampling, all related analyses by statistic physics model is improperly.

We thank the referee for the accurate summary of our work. We believe the comment 1 and 2 are questioning the same point, so we respond to them together here. Indeed, the referee is correct that drawing accurate conclusions about the statistical physics of the protein requires an unbiased sample of particle images.

First, we need to clarify that we did not select particles based on their fine structures. Rather,

a subset of particles (“side view” and “tilt view”), which have high image quality, are selected to reconstruct the fine structure by averaging for 3D classification. This procedure is a standard process in cryo-EM for generating a well-defined high-resolution structure. The subsequent analysis of the conformation pattern statistics presented in the main text was done for this subset of data.

However, in order to make sure we did not introduce bias by selecting particles based on the orientation of their images, we actually did analyze the conformational states of those particles that are not used in the reconstruction stage of our data analysis process (e.g., the “top view” images) together with the particles that are used in the reconstruction. The details of the analysis procedure by including all particles are described in Sec. S2 in the SI with flowcharts of the full analysis process shown as Supplementary Figure 4 and Supplementary Figure 6 for Kai-AA and Kai-EE, respectively in the SI.

Here, for ease of viewing, we reproduced a simplified version of the analysis flowcharts for Kai-AA and KaiC-EE, which are shown below as Figure 1 and Figure 2, respectively. Briefly, particles that are not used in fine structure reconstruction were randomly divided into sub-groups with equal number of particles. Then, each sub-group were combined with the high-quality “side view” and “tilt view” particles to form a combined group. Finally, new 3D reconstruction, classification and statistical analysis are applied to each of the combined groups. The hexamer conformation pattern distribution of particles in two such combined groups are shown in Supplementary Fig.11, whose relevant panels are reproduced as Fig.3 shown below for easy viewing. Besides small quantitative differences, the distribution of hexamer conformation patterns for the combined groups are similar to the hexamer pattern distribution for the group that consists of only high-quality particles, which was reported in Fig. 3a of the main text.

Put together, our results indicate that the choice of the high image quality particles does not introduce bias for the hexamer conformation pattern statistics and including all particles in the analysis does not qualitatively affect our statistical results.

This point has been made clearly at the bottom of page 10 in the revised manuscript with details given in Sec. S2 in the SI.

Fig.1. Flow chart for the processing and statistical analysis of the KaiC-AA “top view” particles that were not used in the fine structure reconstruction.

Fig.2. Flow chart for the processing and statistical analysis of the KaiC-EE “top view” particles that were not used in the fine structure reconstruction.

Fig.3. The distribution of the hexamer conformation patterns for two combined groups (green and red dots) with both “top” view particles as well as high quality (“side view” and “tilt view”) particles and the group with only high quality particles (black dots), which was reported in Fig. 3(a) in the main text.

3. The resolution/map used define the criteria of Bu/Un/Ex conformation is unconvincing, for example, Un 1-7 is similar to Bu 1-12 showed in Supp Fig. 9., It is knowing the resolution of the central portion of particle (such as A-loop) is lower than that near the edge. How the resolution variety affects the criteria of A-loop conformations?

Regarding the specific case mentioned by the referee, the 3D structures of 1-7 and 1-12 are not as similar as they might look like from the projected 2D density map shown in Fig.S9 (b). In particular, the density of Un 1-7 on the bottom right is missing even at the low density threshold (2σ , black mesh). For Bu 1-12, however, the density agrees with the atomic model of the Bu state at the low threshold. This is why the projected 2D density map plot may be misleading and we have to use integral density values (I_1 and I_2) integrated over 3D masks (I_1 and I_2 are defined in the Method section on page 22-23) to distinguish and characterize different conformations. Indeed, the values of I_1 and I_2 for 1-7 and 1-12 are quite different as shown in Fig.S9 (a) – both I_1 and I_2 are larger for 1-12 than those for 1-7.

In general, we agree with the referee that to determine the monomer conformation precisely is quite challenging and we are not aware of any perfect solution to this problem. That is why we introduced an “undefined” state (Un) between the buried (Bu) state and the extended (Ex) state, which serves as a buffer zone to separate the two conformational states. As a result, a wrong call at either of the two decision boundaries (Un-Bu or Un-Ex) would cause a smaller error than mislabeling an Ex-state to be a Bu-state or vice versa.

We have tested the robustness of the pattern distribution with respect to small perturbations near the decision boundary. Here are two examples. In the KaiC-AA dataset, as shown in Fig. 4 below, 1-8 is close to the decision boundary and it was called to be a “Un” state in our study. We now change it to be a ‘Bu’ state, and compare the conformation pattern distribution with the original distribution. As shown in the Fig. 4 (right), the two

distributions are almost identical with the same trend.

Figure 4: (left) Criteria for distinguishing the exposed (Ex) state and buried (Bu) state in KaiC-AA; (right) Comparison of hexamer conformation pattern distributions when 3D volume "1-8" is classified as "Un" or "Bu" state.

In the KaiC-EE dataset, as shown in Fig. 5 below, 2-5 is close to the decision boundary and it was called to be a "Un" state in our study. We now change it to be a 'Bu' state, and compare the conformation pattern distribution with the original distribution. As shown in the Fig. 5 (right) below, the two distributions are almost identical with the same trend. Thus, our main conclusions are not sensitive to the precise position of the classifier boundaries.

Figure 5: (left) Criteria for distinguishing the exposed (Ex) state and buried (Bu) state in KaiC-EE. (right) Comparison of hexamer conformation pattern distributions when 3D volume "2-5" is classified as "Un" (black dots) or "Bu" state (red dots).

We have added the robustness analysis described above to the revised SI (the new Fig. 5c and Fig. 7c in SI).

4. Authors state "Previously reported high-resolution crystal structures with different phosphorylated and phosphomimetic states are nearly identical^{14,15,26,41,42}, indicating that there are functionally important conformational states of KaiC present in solution that

are difficult to capture using crystallography". Author ignored the important crystal structure here (ref; 35, PDB: 3JWQ), in which the crystal structure did showed the conformational change of KaiC by directly binding to KaiB. 3JWQ showed the major conformation induced by binding KaiB is in the loop (Q116-D123) of CI instead of A-loop of CII discovered by cryo-EM average on the mutation sample with statistics. Authors should careful discuss why the conformational change on the CI domain has not been discovered from their mutation sample and selected particles.

From previous biochemical and structural studies, the CI domain has two dominant conformational states discriminated by the hydrolysis of ATP in the CI domain. Compared with 5JWQ, we find our cryoEM structures are both in the pre-hydrolysis state. As KaiC-AA and KaiC-EE both have ATPase activity in the CI domain, we propose that the post-hydrolysis state, and concomitant exposure of the B-loop in CI, is transient. This state maybe stabilized by binding with KaiB, thus we cannot observe this state without KaiB.

5. The directly evidence from the KaiC binding to KaiA and KaiB should be studied to validate the conformational change revealed from the mutation samples.

Previously studies have indicated that mutations at the KaiC phosphorylation sites impact their ability to interact with KaiA and KaiB. KaiC-EE or KaiC-DE strongly interact with KaiB, and KaiC-AA fails to form complexes with KaiB (Nishiwaki et al. 2007, Rust et al. 2011). KaiC-EE has decreased affinity for KaiA relative to KaiC-AA that is correlated with protection of the A-loops from proteolysis (Tseng et al. 2014).

We have now added discussion (with new references) to address comments 4&5 on page 7 in the revised manuscript.

6. Abstract stated the "uncover protein-protein interactions within multi-component biological nanomachines", there is no protein (such as KaiA or KaiB) interaction to the hexamer conformation of KaiC.

Thanks for pointing this out. We have changed the term to "subunit-subunit interaction" throughout.

7. Author stated, "an unsupervised machine learning method" has been used, however, no detailed information about this method, reference, or validation has been described in the manuscript.

Thanks for the point. This method is the RELION algorithm (refs. 1&2) mentioned multiple times in main text. We have clearly stated that now.

Response to reviewer #3's comments

(reviewer comments – black; response – blue; revisions made – red)

Reviewer #3 (Remarks to the Author):

The authors employed cryo-EM and statistical modeling to analyze protein-protein interactions within hexameric KaiC, a protein complex central to maintaining the circadian rhythm in cyanobacteria. The authors identified two main conformation states of KaiC monomers, exposed and buried, and found strong evidence for cooperativity between the conformational states of neighboring monomers within hexamers. The experimental data fits beautifully with the Ising model the authors define in this work, showing that the observed conformational states can be confidently explained by nearest-neighbor subunit cooperativity. Overall, this is a highly original study, and a paradigm for the use of cryo-EM to quantitatively infer protein-protein interactions. That said, the clarity of presentation of some points could be improved, and the analysis of the mixed samples (i.e. composed of monomers of different phosphomimetic mutants) could be made stronger.

We thank the referee for a careful review of our manuscript and positive assessment of our study. In the following, we answer the referee's specific questions and describe the revisions we made in accordance with the referee's suggestions.

Major issues

1. Overall consistency of illustrating “well-resolved” structures:

Given the strategy of the authors' experiments (creating two mutants to mimic two phosphorylation states of KaiC) visual comparative analysis is critical. Figure 1, panels C and D show the same selected secondary structures for, respectively, the KaiC-AA and KaiC-EE phosphomimetic mutants, and this presentation is sufficient for visual comparison. However, panel D also includes additional secondary structures of the A-loop and 422-loop for EE which were not shown for AA. Given the limitation of obtaining the structure of highly dynamic or disordered protein regions, it is not surprising that the more exposed KaiC-AA A-loop was not well-defined in the experimental data. The lack of structure images for the A-loop and 422-loop in KaiC-AA (Figure 1) is seemingly justified in the following paragraph with this argument. However, Figure 2 then appears to show A-loop structures for KaiC-AA. This is inconsistent and confusing. I would expect that either these structures are not well-defined and should not be shown at all, or they are resolved well enough to be shown consistently throughout the manuscript. This point requires clarification.

The referee is correct that both the A-loop and the 422-loop in KaiC-AA are highly dynamic so their densities are not well-defined in the overall average structure (there are both Ex and Bu states in the average structure). To avoid confusion, we have now modified Figure 1(d) in the main text to get rid of the additional secondary structure in KaiC-EE. The revised Figure 1 is shown as Figure 6 below for easy viewing.

Figure 6: The revised Fig. 1 in the main text.

2. Analysis of the mixed sample experiments.

In addition to analyzing hexamers composed exclusively of KaiC-AA or KaiC-EE, the authors also study mixed 50:50 samples of these two mutants. They first measure the correlation function for exposed and buried conformations and fit this to an Ising model, finding a coupling J that is lower than for the pure samples. I would have expected the authors to have then attempted to fit their data by allowing for a reduced J between nearest neighbors of different phosphomimetic states. Indeed, the authors write “...interactions between subunits may depend on their phosphorylation state in addition to their A-loop conformations” and refer to Supplementary Section 3 for incorporation of this dependence. However, this analysis does not appear anywhere in the supplementary information. As it is plausible that the value of J for unlike neighbors will differ from that for like neighbors, this analysis should be performed. Specifically, the authors should report the best fit to the cryo data assuming random arrangements of AA and EE monomers within hexamers, but allowing for a different J between AA and EE monomers, compared to the value of $J = 0.15$ between like monomers found in the pure samples. Even if this approach fails to produce a good fit to the data, the results should be reported, since even a negative result would be informative about how subunits are assorted within hexamers. A related question: Why don't the authors know the arrangement of AA and EE monomers within each hexamer? I would have thought this information might be accessible from the cryo images. Such information would of course greatly inform the model for the mixed sample experiments. I can only assume that this information is not available for some reason, otherwise the authors would have used this important information in their analysis. This point needs to be clarified.

We apologize for the missing information on modeling of the mixed samples, which was

inadvertently lost in revision. Below, we briefly describe the modified Ising model and its result, which we have now mentioned in the main text (bottom of page 15) and included the details in the revised SI.

In this modified model, as the referee correctly guessed, we assumed that the coupling between AA and EE subunits is changed (reduced) by an amount ΔJ from the AA-AA and EE-EE coupling constant J . Mathematically, the modified Hamiltonian can be written as:

$$H_l(\vec{S}) = - \sum_{\langle ij \rangle} J_{ij} S_i S_j - \sum_i [B_{AA}(1 - \sigma_{i,l}) + B_{EE}\sigma_{i,l}] S_i,$$

with $J_{ij} = J - \Delta J \times \frac{(2\sigma_i - 1)(2\sigma_j - 1) - 1}{2}$ and $J=0.14$ is the coupling constant for pure samples.

The monomer arrangements are assumed to be random so each monomer in the ring has an equal probability of being AA or EE, i.e., corresponding to the fully-mixed scenario mentioned in the main text. The best fit for ΔJ is $\Delta J = 0.09 > 0$, which is consistent with the observation of $J_{mix} < J$. However, the improvement in fitting is quite limited with R^2 in this scenario only ~ 0.83 (note that R^2 is as high as 0.99 for the pure samples). This indicates that including a different interaction strength between AA and EE alone cannot explain the experimental data well. This result is now added to the revised SI.

For the second part of the question, the reason that AA and EE monomers cannot be distinguished is because the resolution of our current study is not high enough to distinguish AA and EE. Indeed, this is a very interesting question which we are currently pursuing. This has been clarified explicitly in the main text.

Minor issues

3. Confusing figure - suggested edits for overall clarity (Fig. 1):

- In panels A and B, the hexameric structures of KaiC-AA and KaiC-EE are shown in green/blue and red/yellow respectively. However, within the KaiC-AA structure (panel A) there are small yellow regions between individual monomeric units. Similarly, in panel B there are small blue regions between individual monomeric units. These small colored regions were not defined in the figure caption. Maybe they are the AA and EE residues? Needs clarification.
- It is confusing that Fig. 1 panel D has the A-loop and 422-loop structures alongside the KaiC-EE secondary structures while panel C does not. From my understanding, this is because the A-loop of the KaiC-AA mutant is dynamic and therefore unresolved.
 - o At a minimum, separating the A-loop and 422-loop structures into a distinct panel (i.e., panel E) would help with overall clarity.
 - o The secondary structures of KaiC-AA and KaiC-EE which show the mutation sites (panels C and D left) could be improved by (1) showing in panels A and B where in the hexamer these structures occur (i.e., inside the ring, outside the ring, between monomeric subunits, etc.) and (2) in C and D bolding the A432/A431 and E432/E431 text labels to distinguish them from the other labeled residues which are not mentioned in the main text.

We thank the referee for these thoughtful suggestions to improve the presentation of our figures. We have followed the referee's suggestions and made the following changes:

- In Figure 1, the small yellow regions in panel A and the small blue regions in panel B between individual monomeric units are the 12 ATP molecules bound to the KaiC hexamer.

We have modified the Fig. 1 caption to clarify this point.

- This seems to be the same point as major issue 1. See our response to it before. Briefly, we have revised Fig. 1 (reproduced here as Fig. 6 above) to avoid confusion.

4. Additional statistical calculation to bolster argument (second paragraph, page 9):

- The authors focused their analysis on those hexamers for which all monomers have clearly defined conformational states (24,240 particles for KaiC-AA and 116,785 particles for KaiC-EE). I did a quick calculation starting from the total number of hexamers in each experiment (~1,590,000 for AA and 934,000 for EE) and assuming the occurrence of undefined monomeric conformational states is random and not due to the existence of additional transition states. This yielded that the expected numbers of fully defined hexamers would be ~23,000 for KaiC-AA and ~113,000 for KaiC-EE given the starting number of particles. Given the close agreement of this random expectation with the observed numbers, the natural conclusion is that undefined states are simply randomly unresolved, rather than reflected additional conformational states that are unaccounted for. I suggest the authors redo my calculation more carefully, and if they agree, report this result as supportive of the conclusion that there are only two meaningful monomer conformations.

We thank the referee for bringing our attention to this nice observation. We agree with referee's calculation/estimate, which indeed lend additional support to the fact that the occurrence of Un particles is random and independent, hence it provides a supportive evidence of the conclusion that undefined states are simply randomly unresolved structure and there are only two meaningful monomer conformations.

We have now added details of this calculation in the revised manuscript on page 9.

5. Additional analysis (below equation 1, page 10):

The conformational correlation function was described accounting for the hexamers where at least one of the monomeric subunits had an undefined conformation. What does the correlation function look like only using hexamers with no undefined monomeric subunits?

We compared the correlation functions for two selection criteria: (1) "all pair": all pairs of monomers in which none of the monomers in the pair has an undefined conformation (Un); (2) "no undefined": only using pairs in those hexamers without any undefined monomeric subunits. The comparisons are shown in Fig. 7 below: Fig.7(a) for KaiC-AA; Fig.7(b) for KaiC-EE; Fig. 7(c) for the mixed samples, respectively. These curves do not show significant difference.

Figure 7: Comparison of the correlation functions computed with different selection criteria for (a) KaiC-AA; (b) KaiC-EE; (c) Mixed. The dots are from experiments and the lines are model fits with the correlation length given by ξ 's.

6. Weak comparison statement: (page 12, last paragraph):

- Authors cite Lin et al. (2014) as estimating the subunit free energy difference between the two functional states (buried and exposed) to be 1 ± 0.14 for ST and -1 ± 0.69 for pSpT. From the Ising model the authors describe, they calculate the free energy differences to be 0.38 ± 0.08 for ST mimic (KaiC-AA) and -0.5 ± 0.08 for pSpT mimic (KaiC-EE). Authors state that these values are “consistent with the previous estimates”. While these numbers may have the same sign and relative magnitude, it is somewhat overstated to say they are “consistent”. The authors then detail the “quantitative difference” in the following sentence which contradicts the previous claim that the values are “consistent”. The claim of consistency here should be restated more accurately.

Thanks for the suggestion. The statement is now modified to "are of the same order of magnitude as the previous estimates and their signs and relative values are consistent with the previous estimates."

7. Terminology/wording: (first paragraph, page 9):

- Authors state: “There is a ‘small fraction’ of monomers (25.7% for KaiC-AA and 18.2% for KaiC-EE)...”. A fraction of 26% is not so “small”. Restate more accurately.

Thanks for the suggestion. They are changed simply to "There is a fraction of monomers..".

8. Terminology/wording: (bottom of page 8):

- The term “particle” here refers to the hexameric structure seen on the cryo-EM grids but is not explicitly defined and can therefore be misinterpreted as referring to a monomeric subunit. Without the definition of “particles” equating to the hexamer, it appears highly inconsistent when authors say “...a subset of particles with high quality well-defined structures” were selected and later state “...Un monomers do not have a well-defined structure”. I suggest the authors consistently use the terminology “hexameric particles”.

Thanks for the suggestion. We have revised the manuscript to clearly distinguish the hexameric particle and the monomer subunit.

9. Terminology/wording (second paragraph of page 4):

- Authors state "...whether KaiC monomers in a hexamer interact with each other are unclear". The term "interact" is confusing in this sentence. By definition, individual subunits of a multimer must interact with each other.

We change the "...whether KaiC monomers..." to "...how KaiC monomers...".

10. Confusing figure panel- suggested edit for overall clarity (Fig 2)

- Panel B might be better presented as one structure superposition (rather than three individual ones). Given that SBu1 is identical in each overlay, one simple superposition with the same color legend for each of the three PDB structures would suffice.

We have modified Figure 2 by following the referee's suggestion to make panel B one structure superposition. For ease of viewing, we show the revised Figure 2 below (Figure 8):

Figure 8: The revised Figure 2 in the main text.

11. Confusing figure element- suggested edit for clarity (Fig 3)

- Remove "(pure sample)" from panel C which is confusing and extraneous when not directly comparing this panel to panel B of Figure 4.

We modified Figure 3C in the main text according to referee's suggestion. For ease of viewing, we show the revised Figure 3 below (Figure 9):

Figure 9: Revised Figure 3 in the main text. Statistical results for pure KaiC-AA and KaiC-EE hexamers.

12. Confusing figure element - suggested edit for clarity (Fig 4)

- Remove "(mixed sample)" from panel B which is confusing when not directly comparing this panel to panel C of Figure 3.

We modified Figure 4B in the main text according to referee's suggestion. For ease of viewing, the revised Figure 4 is shown below (Figure 10).

Figure 12: Statistical results for the mixed hexamers.

13. Typo (bottom of page 3):

- “physophorylation” to “phosphorylation”

Thanks for pointing out. It is fixed now.

Response to reviewer #4's comments

(reviewer comments – black; response – blue; revisions made – red)

Reviewer #4 (Remarks to the Author):

The manuscript by Han et al makes important progress towards understanding how the protein nanomachine that comprises the cyanobacterial circadian oscillator is able to keep 24-h time and maintain its observed coherence in the absence of external synchronizing inputs. A wealth of biochemical and structural data over the past 20 years has established the fundamental structures and phosphorylation states that the KaiC hexamer navigates during the cycle, as it complexes with its partners KaiA and KaiB. However, missing from the existing catalog is a measure of the interactions among monomers within KaiC hexamers, and how each is affected by, and/or affects, the phosphorylation state of its neighbors. It has been long recognized that hexamers are mixed with respect to phosphorylation state, but the current data have not addressed either the measurement or implications of such mixtures. The current manuscript addresses the problem by use of Cryo-EM to model the conformations of individual KaiC particles, avoiding the structural constraints that can be imposed by crystallization. The authors use two phosphomimetics that mimic “dawn” and “dusk” states of KaiC and derive key structural features, an exposed and buried state that includes the known A-loop, for each pure hexamer. By classifying mixed hexamers, they are able to conclude that there is significant cooperativity among monomers, which favors neighboring KaiC monomers to be in the same conformational state. The work helps to explain how KaiC mixtures retain coherence even though hexamers assemble and disassemble during the cycle. The work also suggests a mechanism for sharpening the “day-night switch” when KaiC transitions from its phosphorylation to dephosphorylation mode. The work is consistent with prior studies that, in more qualitative or limited ways, predicted the results that were obtained here. This project will inform models that can better explain the mechanism of the clock, whose dynamic structures, rife with allostery and cooperativity, hold the answers to how it tells time. The primary limitation (true of almost all studies on this protein) is that it depends on the phosphomimetic variants of KaiC, which imperfectly reflect KaiC dynamics. However, the current work is a necessary step on the way to understanding how the KaiA, KaiB, KaiC multi-protein mixture changes over time. Overall, this is a very important contribution to understanding the mechanism of the clock.

We thank the referee for a careful review of our manuscript and their positive assessment of our study specially the importance of our work for understanding the mechanism of the circadian clock of Cyanobacteria. Though the reviewer did not have any specific comment for us to address, we have made revisions in our manuscript to improve the presentation of our work in response to the other reviewers' comments.

Reviewers' Comments:

Reviewer #1:

Remarks to the Author:

Dear Editor,

In the revised manuscript titled "Determining Subunit-Subunit Interaction from Statistics of Cryo-EM Images: Observation of Nearest-Neighbor Coupling in a Circadian Clock Protein Complex", Han et al. propose a new approach for identifying subunit-subunit interactions using traditional cryo-EM single-particle analysis and statistical modeling. Although the authors have addressed some feedback from the previous version, several critical points remain unresolved, as detailed below.

Major comments:

1. I disagree with the authors' assertion that the single-particle 3D reconstruction is "not selected particles based on their fine structures. Rather, a subset of particles ('side view' and 'tilt view'), which have high image quality, are selected to reconstruct the fine structure by averaging for 3D classification" (as shown in the response to comment #7). In fact, the classification process in single-particle analysis typically involves selecting a homogeneous population of particles from a heterogeneous pool for 3D reconstruction. The accuracy of this selection and alignment process determines the achieved resolution ("Single-Particle Cryo-EM at Crystallographic Resolution. Cell. 2015 Apr 23;161(3):450-457").
2. The authors' efforts to validate the analysis of conformational states by dividing unused particles into subgroups randomly (as described in Supplementary Fig. 4 and Supplementary Fig. 6) is appreciated. However, these new 3D reconstructions were also obtained via a classification process, in which only a small portion of homogeneous particles (~10%) were selected. Using the same classification procedure to obtain a similar distribution is not surprising. The selected particles, which have a relatively rigid body with a high population of homogeneous structure (corresponding to the ground-state structure), could introduce bias into the statistics. Similar to crystallography, the structural differences between KaiC-AA and KaiC-EE are invisible due to the selection of homogeneous conformations that can form into crystals. Considering the Ising Model and Hamiltonian are established under thermodynamic equilibrium conditions, unbiased sampling is a critical step for the statistics. Therefore, this selection could bias the statistics, and it should be addressed.
3. The U-shaped A-loop should be clearly visible at the claimed resolution of 3.4 Å in Supplementary Fig. 5b and 7b. However, the densities depict the details of the A-loop at the Bu state. It is necessary for the authors to explain why the U-shaped density is not visible at the claimed resolution and how the low-resolution map could be reliable to identify the Ex/Bu states of A-loop.
4. The criteria used to identify the exposed (Ex) state and the buried (Bu) state in the A-loop lack a quantitative definition and solely rely on the visualization of the density. Although the authors introduced an "undefined" state (Un) between the buried (Bu) state and the extended (Ex) state, which serves as a buffer zone to separate the two conformational states. However, it is important to note that the EM density map is derived from the averaged density of particles, and the flexibility of the domain or loop can often cause the smearing of the averaged density, resulting in a lower SNR. The absence of observation of density for the Ex state could also be attributed to a lower percentage of stable structures in the Bu state or greater flexibility of the A-loop within the particles, rather than the existence of a distinct exposed state. In other words, instead of a single state such as an exposed state, it is possible that the A-loop exhibits varying degrees of flexibility and conformational heterogeneity. Classifying all these particles with lower SNR as the Ex state is not accurate. The authors should discuss how this treatment could impact the results.
5. The single particle reconstruction and classification generated 16 maps, which is more than the number of models representing the 13 conformations. This suggests that at least 3 maps should

correspond to the same model. To validate these models, the differences observed among the maps sharing the same model should be analyzed. In other words, the variations in the A-loop density between these maps can be used to assess the diversity of the observed Ex/Bu states. It is essential for the authors to provide a validation analysis of the A-loop density maps among the maps that share the same model.

6. About the sentences of "An interesting observation is that, if we assume the undefined monomers randomly occur in hexamers, the estimated numbers of clearly defined hexamers are $140475 \times (100\% - 25.7\%)! \approx 2.4 \times 10^8$ for KaiC-AA and $371577 \times (100\% - 18.2\%)! \approx 1.1 \times 10^8$ for KaiC-EE, which are in close agreement with the observed numbers", it is unclear where the numbers, 140475 and 371577, come from?

7. The authors claim that the uniqueness of this study lies in the combination of cryo-EM imaging technology, machine learning, and statistical modeling. Actually, both cryo-EM imaging technology and machine learning are standard protocols in single-particle cryo-EM analysis, especially in the latest version of RELION which includes machine learning features. Only the introduction of the Lsing model analysis to the single-particle analysis could indeed be a unique aspect of this study.

Reviewer #3:

Remarks to the Author:

The authors have satisfactorily addressed all of my concerns, and I am happy to recommend publication.

Point-by-point response to Reviewer #1

(reviewer comments in black; our responses in blue; revisions made in red)

In the revised manuscript titled "Determining Subunit-Subunit Interaction from Statistics of Cryo-EM Images: Observation of Nearest-Neighbor Coupling in a Circadian Clock Protein Complex", Han et al. propose a new approach for identifying subunit-subunit interactions using traditional cryo-EM single-particle analysis and statistical modeling. Although the authors have addressed some feedback from the previous version, several critical points remain unresolved, as detailed below.

We thank the reviewer for their continued interest in our work and their specific comments, which we address in the following. Some of the critical points raised by the reviewer lead to revisions of our manuscript to clarify our results, which are also described below.

Major comments:

1. I disagree with the authors' assertion that the single-particle 3D reconstruction is "not selected particles based on their fine structures. Rather, a subset of particles ('side view' and 'tilt view'), which have high image quality, are selected to reconstruct the fine structure by averaging for 3D classification" (as shown in the response to comment #7). In fact, the classification process in single-particle analysis typically involves selecting a homogeneous population of particles from a heterogeneous pool for 3D reconstruction. The accuracy of this selection and alignment process determines the achieved resolution ("Single-Particle Cryo-EM at Crystallographic Resolution. *Cell*. 2015 Apr 23;161(3):450-457").

We thank the referee for this comment, which makes us realize the need to clarify the particle selection process in more details.

As we stated in our previous response, there is indeed particle selection in the RELION software we used in our study. However, the rationale/criterion for particle selection in RELION is to balance the numbers of particles in different projection directions (top-view, tilt-view, side-view) in order to reconstruct the high-resolution 3D structures more accurately. This is a standard practice in 3D reconstruction in cryo-EM image analysis. As stated clearly in ref. 2 ("Processing of Structurally Heterogeneous Cryo-EM Data in RELION. *Methods Enzymol.* 31 May 2016; 579:125-157") by S.H.W. Scheres (the developer of RELION): "One problem that may arise at this point is that there are not enough different views for 3D reconstruction because the particles adopted a strongly preferred orientation on the experimental support. As long as some of the minority views are available in the dataset, throwing away particles from the predominant view may help to balance the orientational distribution and thereby get better reconstructions and classifications" (the second paragraph on page 139 in ref. 2).

In our experiments, there are many more top-view particles than the number of side-view and tilt view particles, which is commonly observed in KaiC cryo-EM studies, e.g., a recent article "Coupling of distant ATPase domains in the circadian clock protein KaiC. *Nat Struct*

Mol Biol. 21 July 2022; 29: 759–766” reported the same observation. The reason for this phenomenon is not clear at present, but we suspect that it may relate to the charge or hydrophilic/hydrophobic properties of protein surface (for example, top view is more hydrophobic and easier exposed to the gas-liquid interface), but this phenomenon should not strongly correlate to the conformational states of the protein. To balance the particle numbers in different orientation, all the non-top-view (tilt-view and side-view) particles (~78,000 for KaiC-AA and ~100,000 for KaiC-EE) are used in our analysis, i.e., no selection, while RELION selected a subset of top-view particles of comparable number as that of the non-top-view particles from a much larger pool of top-view particles (~1,400,000 for KaiC-AA and ~730,000 for KaiC-EE). We have also showed in our paper that the specific choice of which subset of top-view particles to include in our analysis does not change the results significantly (e.g., Supplementary Fig. 12 in SI), see our response to the next comments.

Since this selection process is based on particle orientation, which should not have strong correlation with the fine structure of the KaiC monomer or the spatial configuration of the KaiC monomer around the ring, it does not introduce any bias for the hexamer configuration statistics. To verify this, we have studied the hexamer configuration statistics for the top-view particles and the non-top-view (side-view and tilt-view) particles separately. As shown in Fig. R1 below, the hexamer configuration statistics are the same for the top view particles and the non-top-view particles, which confirm that the particle orientation has no strong correlation with the fine structure of the hexamers for both KaiC-AA and KaiA-EE particles.

Fig. R1. Comparisons of the hexamer configuration statistics for the non-top-view particles only (light blue triangles), the top-view particles only (red triangles), and all particles (black triangles) for (a) Kai-AA particles; (b) Kai-EE particles.

Revisions made: We have now added a new paragraph on page 8-9 to clarify the selection process for 3D reconstruction and the rationale behind it, in particular, why we used only a subset of top-view particles. Furthermore, we have now included another new paragraph on page 11 to describe our test in verifying that particle orientation does not introduce bias in particle fine structure statistics. The figure that describes the comparison of the hexamer configuration statistics for the top-view particles and the non-top-view particles (Fig. R1 above) is now included in the revised SI as the new Supplementary Figure 8.

2. The authors' efforts to validate the analysis of conformational states by dividing unused particles into subgroups randomly (as described in Supplementary Fig. 4 and Supplementary Fig. 6) is appreciated. However, these new 3D reconstructions were also obtained via a classification process, in which only a small portion of homogeneous particles (~10%) were selected. Using the same classification procedure to obtain a similar distribution is not surprising. The selected particles, which have a relatively rigid body with a high population of homogeneous structure (corresponding to the ground-state structure), could introduce bias into the statistics. Similar to crystallography, the structural differences between KaiC-AA and KaiC-EE are invisible due to the selection of homogeneous conformations that can form into crystals. Considering the Ising Model and Hamiltonian are established under thermodynamic equilibrium conditions, unbiased sampling is a critical step for the statistics. Therefore, this selection could bias the statistics, and it should be addressed.

We are glad that our efforts to verify that the selection of a subset of the top-view particles for 3D construction does not affect the hexamer configuration statistics are appreciated by the referee. As we describe in our response to the comment #1, to balance particle orientations, we select a subset of top-view particles while always using all the non-top-view particles in our analysis. To make sure our results do not depend on which subset of top-view particles we chose, we have also used different subsets of top-view particles randomly selected from the remaining pool of the top-view particles together with the same set of non-top-view particles and redid the analysis. As shown in Fig. R2 below (reproduced from Supplementary Fig. 12a&b), our analysis shows that selection of different random subsets of top-view particles from the large top-view particle pool does not change the statistics of the hexamer configurations. The “combined group” shown in Fig. R2 refers to a group of particles that consist of all the non-top-view particles and a random subset of top-view particles of roughly the same number, see Supplementary Fig. 4&6 for details.

Fig.R2. (a) Comparison of the hexamer conformational pattern distribution in main text Fig. 3a (24,240 rings) and that of KaiC-AA combined group 1 (20,133 rings) and combined group 2 (21,638 rings). (b) Comparison of the hexamer conformational pattern distribution in main text Fig. 3b (116,785 rings) and that of KaiC-EE combined group 1 (84,226 rings) and combined group 2 (13,536 rings)). This figure is reproduced from Supplementary Fig. 12a&b in the SI. Note that the statistics are done for all hexamers whose monomers all have a well-defined state (either Bu or Ex).

However, we do not understand what the referee means by “a small portion of homogeneous particles (10%) were selected”. As we described in our paper and in the answer to the referee’s first question (see above), the particle selection is due purely to the need to balance particle numbers in different orientations, which is a standard process in cryo-EM 3D reconstruction, and it has nothing to do with homogeneity of the particles. If one considers particle orientation as an overall characteristic of the particles, our selection process is to increase the heterogeneity of the selected particles in terms of their orientations. For the same reason, we also do not understand what the referee means by “the ground state-structure”. The selection of particles has nothing to do with the energetics of the individual particles. The bottom-line is that the selection process is based on orientation of the particle image in cryo-EM (top-view or side-view, etc.), which has no obvious correlation with the particle’s fine structure or energetics.

We also need to point out that cryo-EM is very different from crystallography, the rapid freezing process puts the biological samples in the near-natural physiological state { “Electron microscopy of frozen water and aqueous solutions. *Journal of Microscopy*. 1982; 128(3): 219-237”}, allowing the dynamic conformational state changes of the protein to be observed.

Revision made: We have added a new paragraph on page 11 wherein we have included additional text to clarify the additional analysis we did for using different subsets of top-view particles and the fact that the statistics of the hexamer configuration patterns does not change significantly with specific choices of the top-view particles.

3. The U-shaped A-loop should be clearly visible at the claimed resolution of 3.4 Å in Supplementary Fig. 5b and 7b. However, the densities depict the details of the A-loop at the Bu state. It is necessary for the authors to explain why the U-shaped density is not visible at the claimed resolution and how the low-resolution map could be reliable to identify the Ex/Bu states of A-loop.

First, we need to point out that the claimed resolution of 3.3 Å is for the overall hexameric ring structure (Main text Fig. 1 on page 6), rather than for every residue in the structure. More importantly, the local structure of the A-loop area is dynamic especially in the Ex-state, which renders its local structure in the A-loop area not as visible as that in the Bu-state, which is more stable. However, the difference between the two conformational states (Ex and Bu) is very clear: for the Ex state, the A-loop is very dynamic thus the Ex state do not have clear electron density; for the Bu state, the A-loop is relatively stable thus have a strong electron density. Furthermore, we have used other independent measures to distinguish the Ex/Bu states (see our response to the next question).

Revision made: We have added a sentence on page 8 to explain why the U-shaped density is not visible for the Ex-state due to its highly dynamic nature.

4. The criteria used to identify the exposed (Ex) state and the buried (Bu) state in the A-loop lack a quantitative definition and solely rely on the visualization of the density. Although the authors introduced an "undefined" state (Un) between the buried (Bu) state and the extended (Ex) state, which serves as a buffer zone to separate the two conformational states. However, it is important to note that the EM density map is derived from the averaged density of particles, and the flexibility of the domain or loop can often cause the smearing of the averaged density, resulting in a lower SNR. The absence of observation of density for the Ex state could also be attributed to a lower percentage of stable structures in the Bu state or greater flexibility of the A-loop within the particles, rather than the existence of a distinct exposed state. In other words, instead of a single state such as an exposed state, it is possible that the A-loop exhibits varying degrees of flexibility and conformational heterogeneity. Classifying all these particles with lower SNR as the Ex state is not accurate. The authors should discuss how this treatment could impact the results.

We would like to point out that the referee's comment -- "The criteria used to identify the exposed (Ex) state and the buried (Bu) state in the A-loop lack a quantitative definition and solely rely on the visualization of the density" -- is inaccurate. Rather than "solely rely on the visualization of the density", our criteria for distinguishing the Ex/Bu state also depend on two quantitative measurements, i.e., the overlap intensities (or the integral density values) $I_{1,n}$ and $I_{2,n}$, as explained in the Main text (Methods-8 on page 21), and shown explicitly in the SM (Supplementary Fig. 5 on page 7; Supplementary Fig. 7 on page 9; Supplementary Fig. 10 on page 12).

We would also like to point out that due to its highly dynamic nature, the Ex state does not have a fixed structure. As we described in the main text, the A-loop in the exposed state tends to stick out of the hexamer and is very dynamic, thus does not have clear electron density. These very flexible conformation states with no obvious density are collectively referred to as "exposed state".

Revision made: We have now added a sentence on the dynamic nature of the Ex-state and its clear definition on page 8.

5. The single particle reconstruction and classification generated 16 maps, which is more than the number of models representing the 13 conformations. This suggests that at least 3 maps should correspond to the same model. To validate these models, the differences observed among the maps sharing the same model should be analyzed. In other words, the variations in the A-loop density between these maps can be used to assess the diversity of the observed Ex/Bu states. It is essential for the authors to provide a validation analysis of the A-loop density maps among the maps that share the same model.

We would like to point out that the configuration pattern and the density map are two different things. The "13 conformational patterns" refers to the 13 different configurations of **KaiC hexamers**, i.e., the 13 different arrangements of the 6 monomers (each in either

Ex or Bu state) around the hexamer ring, as we show clearly in the text and in figures in our paper. For example, “Ex-Ex-Bu-Bu-Ex-Bu” is one such pattern, and “Ex-Ex-Ex-Bu-Bu-Bu” is another pattern. On the other hand, the “16 maps” refers to the 16 high resolution density maps for **KaiC monomers**.

6. About the sentences of “An interesting observation is that, if we assume the undefined monomers randomly occur in hexamers, the estimated numbers of clearly defined hexamers are $140475 \times (100\% - 25.7\%)! \approx 2.4 \times 10^6$ for KaiC-AA and $371577 \times (100\% - 18.2\%)! \approx 1.1 \times 10^7$ for KaiC-EE, which are in close agreement with the observed numbers” , it is unclear where the numbers, 140475 and 371577, come from?

These are the total numbers of particles (with all orientations) used in our analysis presented in the main text (e.g., results shown in Fig. 3) -- KaiC-AA (140,475 particles) and KaiC-EE (371,557 particles).

Revision made: we have clarified the meaning of these numbers in the text on page 10.

7. The authors claim that the uniqueness of this study lies in the combination of cryo-EM imaging technology, machine learning, and statistical modeling. Actually, both cryo-EM imaging technology and machine learning are standard protocols in single-particle cryo-EM analysis, especially in the latest version of RELION which includes machine learning features. Only the introduction of the Lsing model analysis to the single-particle analysis could indeed be a unique aspect of this study.

We agree with the referee that the uniqueness of our study does not lie in any of the individual methods we used in this study but rather the combination of all three. In particular, while cryo-EM and machine learning used in RELION are now standard practice, statistical modeling is necessary to introduce a simple mechanistic explanation (cooperative interactions between nearest neighbor KaiC monomers in the hexamer ring) for the observed statistics in the structure data. We would like to add that accurate statistical modeling is only plausible when there are enough high-quality single particle data, which are obtained through cryo-EM and machine learning based analysis such as RELION.